# Preserving Diversity in Supervised Fine-Tuning of Large Language Models

**Ziniu Li**[1,2]**, Congliang Chen**[1,2]**, Tian Xu**[3]**, Zeyu Qin**[4]**, Jiancong Xiao**[5]**,**
**Zhi-Quan Luo**[1,2]**, and Ruoyu Sun**[1,2,6,†]
[1]The Chinese University of Hong Kong, Shenzhen
[2]Shenzhen Research Institute of Big Data
[3]Nanjing University
[4]Hong Kong University of Science and Technology
[5]University of Pennsylvania
[6]Shenzhen International Center for Industrial and Applied Mathematics
`{ziniuli, congliangchen}@link.cuhk.edu.cn,xut@lamda.nju.edu.cn,`
`zeyu.qin@connect.ust.hk, jcxiao@upenn.edu, {luozq,sunruoyu}@cuhk.edu.cn`

## Abstract

Large Language Models (LLMs) typically rely on Supervised Fine-Tuning (SFT) to specialize in downstream tasks, with the Cross Entropy (CE) loss being the de facto choice. However, CE maximizes the likelihood of observed data without accounting for alternative possibilities. As such, CE usually leads to reduced diversity in the model's outputs, which hinders further development that requires sampling to explore better responses. To address this limitation, this paper introduces a new game-theoretic formulation for SFT. In this framework, an auxiliary variable is introduced to regulate the learning process. We prove that the proposed game-theoretic approach connects to the problem of reverse KL minimization with entropy regularization. This regularization prevents over-memorization of training data and promotes output diversity. To implement this framework, we develop GEM, a new training algorithm that is computationally efficient as CE by leveraging some unique properties of LLMs. Empirical studies of pre-trained models from 3B to 70B parameters show that GEM achieves comparable downstream performance to CE while significantly enhancing output diversity. This increased diversity translates to performance gains in test-time compute scaling for chat and code generation tasks. Moreover, we observe that preserving output diversity has the added benefit of mitigating forgetting, as maintaining diverse outputs encourages models to retain pre-trained knowledge throughout the training process.[1]

## 1 Introduction

Large Language Models (LLMs) (OpenAI, 2023; Dubey et al., 2024) are powerful generative models that excel across specialized tasks. Despite extensive training, pre-trained LLMs often generate tokens without truly understanding users' queries, resulting in irrelevant answers. To enhance performance on specific tasks, researchers employ instruction tuning (Raffel et al., 2020; Wei et al., 2021; Chung et al., 2024), also known as Supervised Fine-Tuning (SFT) (Ouyang et al., 2022; Bai et al., 2022); see Figure 1. This process involves fine-tuning models on high-quality prompt-response pairs, with the Cross-Entropy (CE) loss being the de facto choice for optimization.

However, is CE the most suitable choice for SFT? To answer this question, we must examine SFT's role in the broader context of model development. As the initial post-training stage, SFT teaches LLMs to follow instructions. Once these models can produce responses that are clear, interpretable, and verifiable for annotators, they become well-suited for leveraging human feedback in downstream tasks via techniques like reinforcement learning (Li et al., 2024c). From this perspective, SFT's primary objective is to align the output space of pre-trained LLMs with the preferred formats of downstream tasks. This alignment lays the groundwork for subsequent learning paradigms, including

---

†: Corresponding author.
[1]Code is available at `https://github.com/liziniu/GEM`.

(Rafailov et al., 2023), self-improvement with synthetic data (Adler et al., 2024), and test-time scaling strategies (Snell et al., 2024; Brown et al., 2024; Wu et al., 2024).

Throughout these post-training stages, **output diversity** emerges as a critical factor that enables exploration and identification of higher-quality responses. Beyond its benefits for developers, users also value diversity as it provides multiple options to meet their customized needs.[2] Unfortunately, using CE loss in SFT undermines this goal: while pre-trained LLMs naturally generate diverse outputs (Brown et al., 2020; Wang et al., 2024a), **fine-tuning with CE loss often significantly reduces diversity** (O'Mahony et al., 2024; Kim et al., 2024) (see our own evaluation in Table 2 in the Appendix). This limitation motivates us to reconsider CE's application in LLMs.[3]

Figure 1: Illustration of diversity preservation in SFT. While pre-trained LLMs produce diverse outputs, these often lack proper formatting. Standard SFT using CE improves readability but reduces diversity. We aim to maintain output diversity while enhancing the readability of LLMs' responses.

In theory, CE maximizes the likelihood of the given label, which proved successful for *classification* models (Krizhevsky et al., 2012; He et al., 2016). However, this process inherently overlooks alternative possibilities, fundamentally misaligning with the nature of open-ended *generation* tasks in LLMs, where responses to identical queries can legitimately vary in format, style, or reasoning paths. While this limitation is less problematic during pre-training with massive datasets, it is particularly severe in SFT scenarios where training data is comparatively limited in both size and coverage.

Furthermore, we identify an another key feature of LLMs: they are extensively pre-trained, with rich knowledge encoded in the network. This feature again misaligns with CE during fine-tuning: by suppressing alternative plausible outputs during training, CE can lead to "knowledge forgetting" or "alignment tax", eroding the pre-trained model's encoded knowledge (Ouyang et al., 2022; Luo et al., 2023; Li et al., 2024b). This suggests that "diversity reduction" and "knowledge forgetting" are be deeply interconnected in LLMs. To address these issues, regularization techniques like weight decay (Touvron et al., 2023; Burns et al., 2023) or noisy perturbations to embeddings (Jain et al., 2023) are commonly applied alongside CE loss. However, they have their own limitations (see discussion in Appendix D), highlighting the need for more principled solutions.

**Contributions.** In this paper, we frame the SFT of LLMs as a distribution matching problem. Specifically, we conceptualize the learning process as the transfer of probability mass from source tokens to target tokens within the supervised dataset. To model this, we introduce a game-theoretic framework that incorporates an auxiliary variable (i.e., a meta-controller), which governs the direction and magnitude of probability transfer. This framework enables the development of learning strategies that preserve diversity and mitigate overfitting, as detailed in the main text.

---

[2]AI interfaces like ChatGPT address this need by offering features such as regeneration buttons.

[3]"How to preserve diversity and interestingness" is recognized as an open problem in the talk "ChatGPT and The Art of Post-training" by Barret Zoph and John Schulman (Zoph & Schulman, 2025). We conjecture this diversity-diminishing process is irreversible in post-training and focus on addressing this problem in the first stage (i.e., SFT).

From a theoretical perspective, we demonstrate that our approach connects to a distribution matching problem that involves reverse Kullback-Leibler (KL) divergence minimization with maximum entropy regularization, thereby ensuring the benefits of diversity. Alongside this theoretical insight, we develop a practical training algorithm, GEM, for the game. GEM leverages some unique properties of LLMs, offering computational efficiency and scalability comparable to optimizing the CE loss.

We empirically validate the effectiveness of GEM, by fine-tuning pre-trained models ranging from 3B to 70B in size. We have two main findings. First, GEM enhances the model's ability to generate more diverse responses, which translates into improved performance in test-time scaling. For example, when fine-tuning Llama-3.1-8B (Dubey et al., 2024), GEM outperforms CE by achieving a 5-point improvement in chatting and an 8-point improvement in code generation tasks through repeated sampling. Importantly, to match the performance of baselines, GEM often requires only 0.5x the sampling budget. Second, by preserving output diversity, GEM also mitigates forgetting, demonstrating an 83% reduction in alignment tax.

To summarize, our contributions are threefold:

- We introduce a game-theoretic framework for SFT, enabling a controlled design to preserve diversity. Specifically, we theoretically prove that it can achieve the distribution matching with reverse KL minimization and maximum entropy regularization.
- We develop a new training method GEM that effectively solves the game in practice. This algorithm offers computational efficiency advantages over previous methods designed for similar games.
- We empirically validate that preserving diversity in distribution matching enhances test-time scaling performance while reducing forgetting and alignment tax.

## 2 PRELIMINARY

**Large Language Models.** Let $f$ be the generative distribution modeled by the language model, where $f(y|x)$ denotes the conditional distribution of response $y$ given the prompt $x$. Typically, $f$ is parameterized by a Transformer (Vaswani et al., 2017), with parameters $\theta$. A key feature of $f$ is that the generation space $\mathcal{Y}$ is finite, which enables favorable optimization properties that we will discuss later. Note that $y$ and $x$ may be sequential, in which case an auto-regressive formulation is employed.

**Supervised Fine-Tuning and the Cross-Entropy Method.** To specialize in downstream tasks, LLM relies on SFT after pre-training. This process involves using a supervised dataset with high-quality prompt-response pairs $(x, y)$, sampled from the prompt distribution $\rho$ and conditional data distribution $p(y|x)$. The Cross Entropy (CE) loss is the de facto training objective for SFT, designed to maximize the likelihood of the training data. Formally, this is expressed as:

$$\min_\theta \mathcal{L}_{\mathrm{CE}}(\theta) = -\mathbb{E}_{x \sim \rho}\mathbb{E}_{y \sim p(\cdot|x)}[\log f_\theta(y|x)].$$

Here, the prompt distribution $\rho$ is typically not modeled during SFT and can be treated as a constant; for simplicity, we omit it when the context is clear. A key feature of this approach is that it exclusively maximizes the likelihood of the observed data, disregarding alternative plausible responses. Please refer to Figure 10 in the Appendix for the theoretical understanding of CE.

## 3 CHALLENGES AND PRINCIPLES FOR SFT

Before exploring technical solutions, we establish guiding principles for SFT. We note that SFT is rarely the final stage of LLM development; subsequent phases such as preference learning (Rafailov et al., 2023; Azar et al., 2024; Wang et al., 2024b), reinforcement learning (Li et al., 2024c; Shao et al., 2024), and advanced inference-time strategies (Snell et al., 2024) heavily depend on sampling and output **diversity** to explore high-quality solutions. This reliance on diversity underscores a key challenge: while pre-trained LLMs produce diverse outputs due to their broad knowledge bases, standard SFT practices—particularly the use of CE loss—often reduces this diversity (O'Mahony et al., 2024; Wang et al., 2024a). Such reduction can lead to knowledge forgetting, aligning with the "alignment tax" phenomenon observed in (Bai et al., 2022; Ouyang et al., 2022).

We argue that preserving output diversity during SFT can address these issues. Intuitively, the ability of a model to generate diverse responses serves as an indicator of the richness of its retained knowledge. By maintaining diversity, the model is compelled to consider alternative plausible responses, which in turn necessitates that its internal parameters encode and retain relevant knowledge. To operationalize

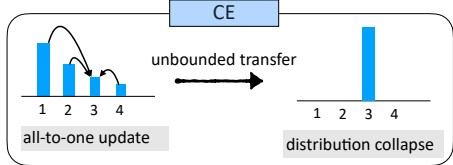 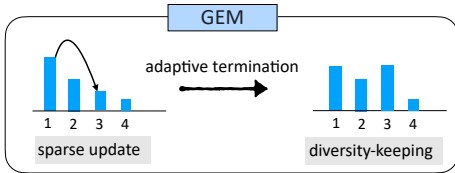

Figure 2: Comparison of learning schemes: CE v.s. GEM ($\beta = 0$). The arrows illustrate the probability movement directions during the learning process, with Token 3 as the target token.

this insight, we propose the following guiding principle for SFT: *learning from the data while preserving diversity.*[4] In the following sections, we present technical insights and solutions.

## 4  PROBABILITY TRANSFER THEORY

In this section, we draw insights into algorithm design by examining the dynamics of CE. We will introduce a new theory of probability transfer. To illustrate this concept, we study a simplified setting, where the prompt $x \in \mathcal{X}$ is fixed and given. We model the distribution $f_\theta(y|x) = \texttt{softmax}(\theta_x)$ with $\theta_x \in \mathbb{R}^K$ being the "logit" and $K$ being the vocabulary size. Let $y = i \in [K]$ denote the token class to be learned. We begin by calculating the gradient of the CE loss for the given example:

$$-\nabla_\theta \mathcal{L}_{\text{CE}}(\theta) = [-f_\theta(1|x), -f_\theta(2|x), \dots, 1 - f_\theta(i|x), \dots, -f_\theta(K|x)]. \tag{1}$$

This indicates that, except for the label class $i$, whose logit increases by $1 - f_\theta(i|x)$, all other tokens experience a logit decrease proportional to their probabilities.

What makes this behavior particularly interesting? We interpret it through a logit flow dynamics perspective, where logits are redistributed among token classes during training. Let $i$-th token class being the "target", while other tokens being the "sources". We have the following observation.

**Proposition 1.** *The gradient of CE specifies a logit flow map: each source token $j$ transfers $f_\theta(j|x)$ logits to the target token $i$. Formally,*

$$-\nabla_\theta \mathcal{L}_{\text{CE}}(\theta) = \sum_{j=1, j \neq i}^{K} w_{i \leftarrow j} \cdot e_{i \leftarrow j} \tag{2}$$

$$w_{i \leftarrow j} = f_\theta(j|x)$$

$$e_{i \leftarrow j} = [0 \ \cdots \ \underbrace{1}_{i\text{-th position}} \ \cdots \ \underbrace{-1}_{j\text{-th position}} \ \cdots \ 0]$$

*where $w_{i \leftarrow j}$ acts as weighting factor, and $e_{i \leftarrow j}$ is a vector with $i$-th element being $1$ and the $j$-th element being $-1$ and $0$ otherwise. Furthermore, the logit flow satisfies a conservation property, ensuring that logits redistributed from source tokens equal those received by the target token:*

$$\textit{Logits from source tokens} = \sum_{j=1, j \neq i}^{K} f_\theta(j|x)$$

$$= 1 - f_\theta(i|x) = \textit{Logits to the target token}.$$

Here, we provide an example for numerical illustration. Consider $f_\theta = [0.4, 0.3, 0.2, 0.1]$ with label $i = 3$. The CE gradient decomposes as:

$$-\nabla_\theta \mathcal{L}_{\text{CE}}(\theta) = [-0.4, -0.3, 0.8, -0.1]$$

$$= 0.4 \times [-1, 0, 0, 1] + 0.3 \times [0, -1, 1, 0] + 0.1 \times [0, 0, 1, -1].$$

That is, there are 3 flows: tokens 1, 2, 4 act as sources, transferring logits to the target 3. The theorem provides insight into CE's optimization dynamics for a single gradient step. We formalize CE's iterative optimization procedure in Figure 3.

We highlight CE's limitations and present our proposed techniques below.

---

[4]While preserving diversity may retain suboptimal responses, we defer this issue to the RL stage, where the model can refine its outputs through trial-and-error, leveraging feedback from reward mechanisms.

(CE)

> While there exists source token $j \neq i$ with $f_{\theta_k}(j|x) > 0$, continue the following steps.
> - Find any $j$ with $f_{\theta_k}(j|x) > 0$
> - Decrease the logit for source token $j$ by learning rate $\eta$ and weight $w_{i \leftarrow j}$:
> $$\theta_{k+1}[j] = \theta_k[j] - \eta * w_{i \leftarrow j}$$
> - Increase the logit for the target token $i$ in a similar manner:
> $$\theta_{k+1}[i] = \theta_k[i] + \eta * w_{i \leftarrow j}$$

Figure 3: CE's optimization procedure in the probability transfer viewpoint.

**Limitation 1 of CE: All-to-One Probability Transfer.** CE loss implements a logit flow mechanism where probability mass from all non-target tokens is transferred to the target token (c.f. the first bullet point in Figure 3). This approach penalizes all non-target tokens regardless of their semantic relevance or contextual appropriateness. For example, in the sentence "I like coffee", while "coffee" is the target, reducing the logit and probability of "tea"—a semantically related and contextually plausible token—may harm generalization. This limitation is especially critical in LLMs, as they are extensively pre-trained and encode rich knowledge across many tokens. The all-to-one flow dynamic disrupts these carefully learned token relationships, potentially reducing output diversity and contributing to knowledge forgetting.

**Proposed Technique 1: Sparse Update.** To address this limitation, we propose a sparse update strategy. Instead of considering all source tokens, we select only pivotal tokens for probability transfer (c.f. the first bullet in Figure 4). Specifically, we introduce a simple approach: we identify the pivotal token as the one with the highest model confidence: $j \in \arg\max f(\cdot|x)$. The underlying intuition is clear: the model only corrects its most confident prediction if it is incorrect (i.e., does not match the target label). This targeted correction mechanism protects the probability distribution of minority cases, ensuring they maintain a reasonable chance of being sampled during generation.

However, this technique alone is insufficient. As training progresses and the pivotal token's probability diminishes through repeated updates, the flow mechanism naturally shifts to other high-probability tokens. This cascading effect eventually approaches the dense update behavior of CE as probabilities of all source tokens are sequentially reduced. This observation highlights the need for a principled approach to terminate probability transfer at an appropriate stage.

**Limitation 2 of CE: Unbounded Probability Transfer.** The CE optimization process lacks a natural termination point for probability transfer (c.f. the first line in Figure 3). The logit flow continues indefinitely until all source token probabilities approach zero, causing the distribution to collapse and concentrate entirely on the target token. This represents an undesirable convergence point that eliminates distribution diversity. Fundamentally, this issue stems from CE's implicit assumption that observed data should be assigned maximum likelihood, without preserving reasonable probability for alternative tokens.

**Proposed Technique 2: Adaptive Termination.** To prevent distribution collapse, we introduce an intuitive stopping criterion: halt the probability transfer once the target token $i$ becomes the most probable token in the distribution (c.f. the first line in Figure 4). Formally, if $i \in \arg\max f(\cdot|x)$, we stop further update. This supports the assumption that while the observed data should be adjusted to increase its likelihood, other possibilities should still be considered, so the probability of the observed data should not be forced to 1. Our stopping rule is designed to ensure that, after learning, greedy decoding (i.e., selecting the token with the highest probability) can output the correct label, which achieves the primary training objective. An additional advantage of this rule is that, due to early termination, it keeps the resultant distribution close to its initial state, thereby mitigating forgetting.[5]

The two techniques outlined above form the foundation for the initial prototype design of our proposed algorithm, GEM (Game-theoretic Entropy Maximization); see Figure 2 and Figure 4. The meaning and rationale behind the name will be explained in detail in Section 5.

---

[5]This serves as a form of implicit regularization, encouraging the model to remain close to its pre-trained state. Unlike heuristic approaches that merely keep the model close to its initial parameters—which inevitably face a trade-off between performance and closeness—our design identifies a clear constraint: ensuring greedy decoding can output the information to be learned. Within this constraint, we maximize closeness to the pre-trained model.

(GEM) 

While the target token $i \notin \arg\max f_{\theta_k}(\cdot|x)$, continue the following steps.

- Calculate the model's best prediction $j = \arg\max f(\cdot|x)$
- Decrease the logit for source token $j$ by learning rate $\eta$ and weight $w_{i \leftarrow j}$:
$$\theta_{k+1}[j] = \theta_k[j] - \eta * w_{i \leftarrow j}$$
- Increase the logit for the target token $i$ in a similar manner:
$$\theta_{k+1}[i] = \theta_k[i] + \eta * w_{i \leftarrow j}$$

Figure 4: GEM's optimization procedure ($\beta = 0$) in the probability transfer viewpoint.

While this prototype is conceptually sound and aligns with the guiding principle of SFT, it lacks the flexibility needed to extend its ideas to neural network training scenarios. To address this limitation, we have developed a more general mathematical framework that refines the approach, making it both more elegant and adaptable. We will discuss this framework in detail in the next section.

## 5 GAME-THEORETIC FORMULATION

In this section, we present a mathematical framework for fine-tuning approaches that preserve diversity. We begin by highlighting a key observation from Section 4: introducing a meta-level mechanism helps govern the the logit flow, such as determining sparse updates and defining stopping rules. Mathematically, this corresponds to the usage of an auxiliary variable. This insight motivates the development of a framework that integrates such an auxiliary variable. We find that a game-theoretic framework (Goodfellow et al., 2014; Jolicoeur-Martineau, 2019) is well-suited for this purpose, as it naturally introduces an additional player.

Specifically, our proposed mathematical framework is formally presented as follows:

$$\min_f \quad \mathcal{L}_{\text{GEM}}(f, q) \triangleq \mathbb{E}_x \mathbb{E}_{y^{\text{real}} \sim p(\cdot|x)} \mathbb{E}_{y^{\text{gene}} \sim q(\cdot|x)} \left[ \log f(y^{\text{gene}}|x) - \log f(y^{\text{real}}|x) \right] \quad (3)$$

$$\max_q \quad \mathcal{Q}(f, q) \triangleq \mathbb{E}_x \mathbb{E}_{y^{\text{gene}} \sim q(\cdot|x)} \left[ \log f(y^{\text{gene}}|x) \right] + \beta \cdot \mathcal{H}(q(\cdot|x)). \quad (4)$$

In this framework, the term $y^{\text{real}}$ represents the supervised label in the dataset, while $y^{\text{gene}}$ corresponds to the model-generated output. For clarity, $y^{\text{real}}$ aligns with the target token and $y^{\text{gene}}$ aligns with the source token in the probability transfer framework. The symbol $q$ refers to a distribution similar to $f$. Notably, $q$ serves as the auxiliary variable introduced to regulate the process, achieving the functions of sparse updates and proper termination introduced in Section 4.

A simple explanation of our framework is that we aim to maximize the log-probability $f$ for real data while minimizing the log-probability probability for model-generated data, as expressed in Equation (3). Simultaneously, we optimize the auxiliary variable $q$ to enhance its performance by treating $\log f$ as a objective function. However, this overview omits key insights into the underlying approach. In the following section, we will provide a more detailed explanation of our framework.

### 5.1 UNDERSTANDING THE GAME

In this section, we explain the game-theoretic framework and connect it to the probability transfer theory developed in Section 4. For clarity, we adopt the same setting as in Section 4 and calculate the gradient with respect to $\theta$. For a specific sample where $y^{\text{real}} = i$, the gradient is given by:

$$-\nabla_\theta \mathcal{L}_{\text{GEM}}(f_\theta, q) = \sum_{j=1, j \neq i}^{K} w_{i \leftarrow j} \cdot e_{i \leftarrow j}, \quad (5)$$

$$w_{i \leftarrow j} = q(j|x).$$

The form is similar to the CE flow in Equation (2), except that the weights are now design variables governed by $q$, introducing greater flexibility. We will explore the design of $q$ and show that this framework generalizes both the CE formulation and the GEM prototype discussed in Section 4.

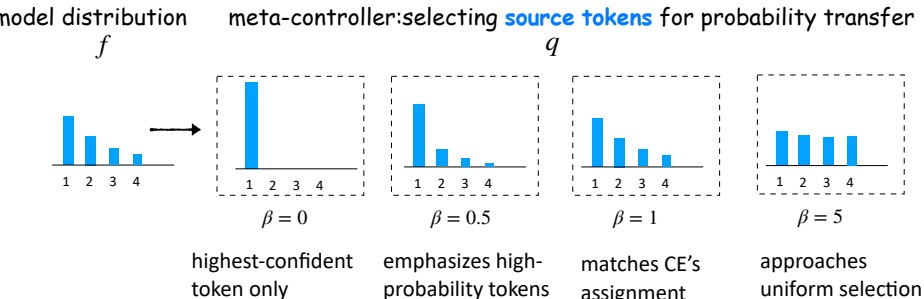

Figure 5: Illustration of the meta-controller $q$.

Our design of $q$ follows an optimization problem in Equation (4):

$$\underset{q}{\operatorname{argmax}} \, \mathcal{Q}(q, f) = \begin{cases} \delta_j(x) \text{ with } j = \operatorname{argmax} f_i(\cdot|x) & \text{if } \beta = 0 \\ \texttt{softmax}(1/\beta * \log f(y|x)) & \text{if } \beta > 0 \end{cases} \tag{6}$$

That is, $q$ is a shifted distribution derived from $f$. This transformation is visualized in Figure 5.

Note that $q$ serves as a "controller" for prioritizing source tokens. It determines both the selection of source tokens and the strength of their associated flows. When $\beta = 0$, $q = \delta_j(x)$ (i.e., the Dirac distribution) selects the single token with the highest probability in $f$, which corresponds exactly to the algorithm prototype in Section 4. When $\beta = 1$, $q$ becomes identical to $f$, reducing to the CE formulation. For intermediate values $\beta \in (0, 1)$, $q$ represents a soften distribution where high-probability modes become more prominent, while low-probability regions are further suppressed. This results in reduced contribution from minority tokens in the probability transfer, thereby protecting their probabilities. Thus, for small $\beta$ values, the framework encourages sparse updates and safeguards tokens that may not appear in the current dataset but were learned during pre-training. When $\beta > 1$, the framework encourages leveraging minority tokens for probability transfer, diminishing diversity more significantly—an approach not considered in our framework.

A key feature of $q$ is its adaptability. Rather than implementing a uniform controller for all prompts, $q$ adapts to each distribution's unique characteristics. This adaptability is particularly well-suited for LLM training, where prompts come from diverse tasks and distribution shapes vary considerably across different prompts, similiar to the challenge highlighted in (Song et al., 2023; Li et al., 2024c).

## 5.2 THEORETICAL GUARANTEE

Building on our earlier illustrative analysis with a single data point, we now formally present a theory with the real data distribution $p$.

**Proposition 2.** *For a data distribution satisfying $p(y|x) > 0$, with $\beta > 0$, the game in Equations (3) and (4) posses a unique Nash equilibrium point:*

$$\begin{cases} f^\star = softmax(\beta * \log p) \\ q^\star = p \end{cases} \tag{7}$$

*Furthermore, $f^\star$ corresponds to the optimal solution to the distribution matching problem (with $1/\beta = (\gamma + 1)$), which minimizes the reverse KL divergence with entropy regularization:*

$$f^\star = \underset{f}{\operatorname{argmin}} \, \mathbb{E}_x \left[ D_{\mathrm{KL}}(f(\cdot|x), p(\cdot|x)) - \gamma \mathcal{H}(f(\cdot|x)) \right]. \tag{8}$$

This result provides an intuitive understanding of the distribution matching problem that our framework addresses: it drives the distribution $f$ close to $p$ while encouraging the diversity through entropy regularization. This is exactly the goal we set in Section 3. We note that our analysis relies on $\beta > 0$, as the equilibrium point is unique in this scenario. For $\beta = 0$ and $p$ is the Dirac distribution, we can still apply the stopping criterion discussed in Section 4. However, the choice of $f^\star$ is neither unique nor analytically tractable, introducing additional complexities that we leave for future work.

## 5.3 TRAINING ALGORITHM: GEM

In this section, we present a algorithm for training neural networks within our framework. For clarity, in the main text, we state the token-level version, where $f(y|x)$ denotes the one-step conditional

distribution, while the extension to the sequence-level is provided in Appendix B. Specifically, we parameterize $f_\theta$ using a Transformer and optimize its parameter $\theta$ directly. The overall algorithm is outlined in Algorithm 1, which incorporates two key features: single-model optimization and variance-reduced gradient estimation. These features ensure that our algorithm is highly scalable, requiring nearly the same GPU memory and computational speed as optimizing the standard CE loss.

**Single-Model Optimization.** Recall that we have a closed-form solution for $q$ (see Equation (4)), which significantly simplifies the training procedure. Specifically, the update rules are as follows:

$$
\begin{cases}
f_{\theta_{k+1}} = f_{\theta_k} - \nabla_\theta \mathcal{L}_{\text{GEM}}(f_\theta, q_k) \mid_{\theta=\theta_k} \\
q_{k+1} = \arg\max_q \mathcal{Q}(f_{\theta_{k+1}}, q) = \texttt{softmax}(1/\beta * \log f_{\theta_{k+1}})
\end{cases}
$$

Specifically, $q$ can be computed simply by shifting the logits of $f_\theta$, eliminating the need to maintain a separate network for $q$. This reduces the memory burden of training. In contrast, GANs require an additional neural network (i.e., the discriminator), which must be trained alongside the main model.

**Variance-reduced Gradient Estimation.** Building on the above observation, we slightly adapt the notation to define the loss function. Let $\theta$ denote the parameters of the distribution $f$. Given training samples $(x_i, y_i^{\texttt{real}})$, the loss is defined as (with slight notation abuse):

$$
\mathcal{L}_{\text{GEM}}(\theta) = \sum_i \sum_{y^{\text{gene}}} q_k(y^{\text{gene}}|x_i) \cdot \left[\log f_\theta(y^{\text{gene}}|x_i) - \log f_\theta(y_i^{\texttt{real}}|x_i)\right].
$$

A notable feature is that it computes the *true* expectation over $q_k$. This reduces the variance of the gradient estimator and improves the training stability. This again differs from GANs, where stochastic gradient estimation introduces randomness from both the data distribution $p$ and the generated distribution $q$. We note that these optimization properties arise from the fact that distributions in LLMs have finite support, while GANs typically operate in continuous domains.

---

**Algorithm 1** GEM

---

**Input:** Dataset $\mathcal{D} = \{(x_i, y_i^{\texttt{real}})\}$
1: **for** iteration $k = 1, \dots, K$ **do**
2:      Set $q_k = \texttt{softmax}(1/\beta * \log f_{\theta_k})$
3:      Loss $\mathcal{L}_{\text{GEM}}(\theta) = \sum_i \sum_{y^{\text{gene}}} q_k(y^{\text{gene}}|x_i) \cdot \left[\log f_\theta(y^{\text{gene}}|x_i) - \log f_\theta(y_i^{\texttt{real}}|x_i)\right]$
4:      Update $\theta_{k+1} = \theta_k - \eta \cdot \nabla_\theta \mathcal{L}_{\text{GEM}}(\theta) \mid_{\theta=\theta_k}$
**Output:** Generative model $f_{\theta_{K+1}}$

---

## 6 EXPERIMENTS

In this section, we present experiment results validating the effectiveness of our proposed framework, demonstrating that GEM matches the downstream performance of CE while offering two distinct advantages: (1) its diversity preservation enables a wider range of outputs, enhancing test-time scaling performance, and (2) it mitigates forgetting, effectively reducing the alignment tax. Extended results are available in Appendix F.

**Set-up.** We fine-tune the pre-trained Llama-3.1-8B model with the `UltraFeedback` dataset (Cui et al., 2024). This dataset contains prompts from instruction datasets like Evol-Instruct and UltraChat, and responses generated by models such as GPT-4 and Llama-2-7B/13B/70B-Chat. Following (Yu et al., 2023; Liu et al., 2023; Cui et al., 2024), we set the learning rate to $2 \times 10^{-5}$, employing a cosine learning rate decay schedule, and use a macro batch size of 128. The maximum sequence length, encompassing both the prompt and response, is set to 2,048 tokens. Models are trained for three epochs. Detailed experimental settings are described in Appendix E.

We implement GEM with $\beta = 0.7$. Our primary baseline is the standard CE loss. Additionally, we explore a variant of CE incorporating a weight decay of 0.1, which has been commonly used in previous studies (Ouyang et al., 2022; Bai et al., 2022). We refer to this approach as CE + WD. The NEFT method (Jain et al., 2023), which perturbs the input embedding with random noise in fine-tuning to mitigate overfitting, has also been implemented.

### 6.1 IMPROVING DIVERSITY AND TEST-TIME SCALING

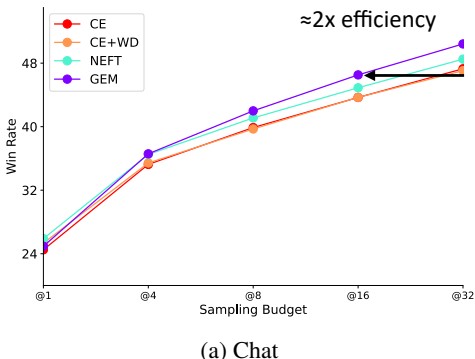 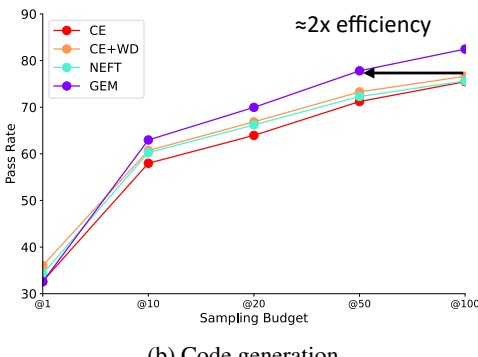

(a) Chat

(b) Code generation

Figure 7: Performance of test-time scaling. The results demonstrate that GEM achieves better performance with the same sampling budget and is more efficient in reaching comparable performance.

In this section, we demonstrate that through implicit entropy maximization, GEM enhances output diversity. This is evident from the entropy values of the generative distribution: 0.42 (CE), 0.41 (CE + WD), 0.43 (NEFT), and 0.76 (GEM). Consequently, GEM is more likely to sample diverse solutions and has a higher chance of identifying better solutions during inference through repeated sampling. This aligns with the recent trend of test-time scaling (Snell et al., 2024; Brown et al., 2024; Wu et al., 2024). We illustrate this benefit using two tasks: chat and code generation, while mathematical reasoning tasks are explored in Appendix F.

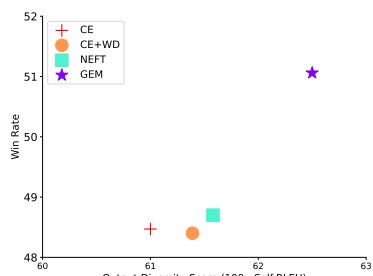

Figure 6: Enhancing output diversity boosts the win rate when using BoN.

**Chat.** We assess the model's chat ability using the best-of-N sampling strategy. We prompt the trained models to answer 805 questions from the AlpacaEval dataset (Li et al., 2023). For each question, the model generates 32 responses, from which a reward model selects the best response. We employ the reward model FsfairX-LLaMA3-RM-v0.1[6], which has top performance on RewardBench (Lambert et al., 2024). The selected response is compared with GPT-4's response with the win rate reported in Figure 7 (a).

The evaluation reveals that weight decay does not lead to performance improvements. On the other hand, NEFT shows strong performance, partly attributed to its longer responses, as highlighted in (Jain et al., 2023). However, GEM outperforms CE, with a 3.1-point (6.6% relative) increase in win rate. Additionally, we note that GEM achieves comparable performance to CE while requiring about 2x fewer sampling costs. To further understand this improvement, we evaluate output diversity using the Self-BLUE metric[7] across generated responses, as depicted in Figure 6. GEM not only increases output diversity but also enables the generation of higher-quality responses.

**Code Generation.** We consider the HumanEval (Chen et al., 2021) benchmark, in which the model is asked to generate Python codes for 163 questions, and the executor judges their correctness. The evaluation metric is the pass rate among generated responses. The results are reported in Figure 7 (b).

In this task, we observe that for pass@100, weight decay slightly improves the performance of CE, increasing it from 75.5 to 76.6, while NEFT shows no significant improvement, maintaining a performance of 75.6. In contrast, GEM achieves a 7-point improvement, reaching a performance of 82.5, which represents a 9.3% relative increase. Furthermore, GEM proves to be highly efficient in test-time scaling, requiring only about half the sampling budget to achieve comparable performance.

We note that in our experiments, the performance using BON or pass rate serves as an estimate of self-improvement. Specifically, high-quality self-generated samples produced by GEM can be distilled back into the model, enhancing its zero-shot performance (see (Sessa et al., 2024)). Therefore, we believe GEM can exhibit a more effective scaling in post-training, a topic we plan to explore in future

---

[6]https://huggingface.co/sfairXC/FsfairX-LLaMA3-RM-v0.1

[7]Self-BLUE measures similarity among responses; so, we use 100 - Self-BLUE as a metric for diversity.

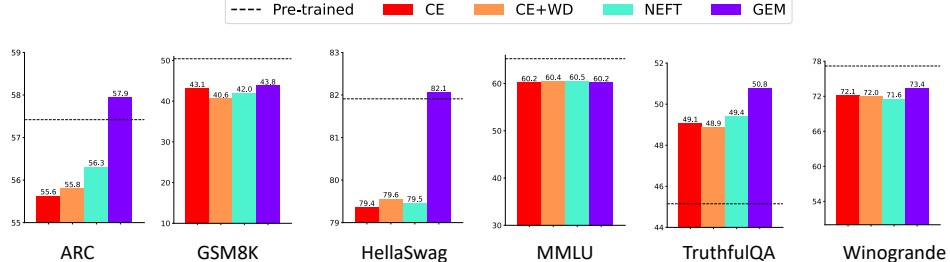

Figure 8: Performance on tasks from the `OpenLLM leaderboard`. The results indicate that GEM outperforms CE, demonstrating a lower alignment tax.

work. We also note that in the above evaluation, GEM enhances output diversity without sacrificing its direct generation performance: in chat, CE (24.5) vs. GEM (24.9), and in code generation, CE (32.7) vs. GEM (32.6). This contrasts with other techniques, which often increase diversity at the cost of performance degradation; see Appendix D for further discussion.

## 6.2 MITIGATING FORGETTING AND REDUCING ALIGNMENT TAX

In this section, we show that GEM also helps mitigate forgetting and reduce alignment tax. We evaluate the models across six tasks: ARC, GSM8K, HellaSwag, MMLU, TruthfulQA, and WinoGrande, as listed on the `OpenLLM leaderboard`. Results are presented in Figure 8.

We observe that after SFT, performance declines on most tasks for the baseline models, with CE showing the most significant drops. However, GEM exhibits different behavior: for tasks such as ARC and HellaSwag, performance does not decrease. On average across six tasks, GEM shows an **80% reduction in alignment tax**, with a 0.3-point drop for GEM compared to a 1.5-point drop for CE. This behavior aligns with our method, which introduces $q$ to encourage sparse updates of tokens and prevent forgetting. Further

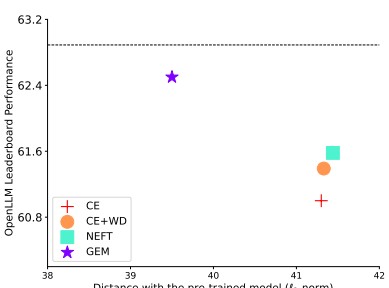

Figure 9: Staying close to the pre-trained model helps mitigate forgetting.

validation can be seen by examining the distance from the pre-trained model. Specifically, we measure the parameter difference from the initialization using the $\ell_2$-norm, as shown in Figure 9. The results indicate that GEM is closer to the pre-trained model in parameter space, leading to an interesting conclusion that diversity-preserving updates help reduce the model's deviations during learning.

## 7 CONCLUSION

In this paper, we identify a key limitation of SFT via optimizing the CE loss: its tendency to encourage over-memorization of training data and the resultant reduction in output diversity, which could hinder subsequent development that relies on sampling to explore better responses. To address these limitations, we develop a new game-theoretic formulation. Central to this framework is a meta-controller that encourages sparse updates and ensures proper termination of the learning process. We validate the advantages of the proposed framework in test-time scaling and its effectiveness in mitigating catastrophic forgetting and reducing alignment tax.

We focus on SFT in this paper, but our techniques may also be of interest for developing algorithms to preserve diversity and mitigate forgetting. In particular, the enhanced diversity achieved by our approach can be beneficial in several contexts: it enables exploration to improve performance limits with reinforcement learning (Li, 2025), facilitates self-distillation with best-of-n sampling (Sessa et al., 2024), and enhances the diversity of synthetic data generation (Adler et al., 2024). Please see Appendix G for the discussion. We see the potential of our method in these areas and plan to explore these topics in future work.

## ACKNOWLEDGMENT

Ziniu Li would like to thank Yao Fu for the discussion about diversity during the ICML 2024 poster session, as well as Yushun Zhang for reading the manuscript and providing valuable feedback. The work of Ruoyu Sun was supported by NSFC (No. 12326608); Hetao Shenzhen-Hong Kong Science and Technology Innovation Cooperation Zone Project (No. HZQSWS-KCCYB-2024016); University Development Fund UDF01001491, the Chinese University of Hong Kong, Shenzhen; Guangdong Provincial Key Laboratory of Mathematical Foundations for Artificial Intelligence (2023B1212010001). The work of Z.-Q. Luo was supported by the Guangdong Major Project of Basic and Applied Basic Research (No.2023B0303000001), the Guangdong Provincial Key Laboratory of Big Data Computing, and the National Key Research and Development Project under grant 2022YFA1003900. The work of Tian Xu was supported by the Fundamental Research Program for Young Scholars (PhD Candidates) of the National Science Foundation of China (623B2049).

## ETHICS STATEMENT

Our work focuses on designing better algorithms for fine-tuning large language models, aiming to enhance their effectiveness and broaden their applications. In particular, the entropy regularizer we introduce for distribution matching encourages more diverse outputs from language models. We do not foresee any direct negative impacts from this approach.

## REPRODUCIBILITY STATEMENT

The proof of Proposition 2 is provided in Appendix C. Experiment details for reproducing our numerical results can be found in Appendix E. Our code is accessible at https://github.com/liziniu/GEM.

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

## A  RELATED WORK

**Supervised Fine-Tuning.** The phenomenon of alignment tax in fine-tuning LLMs was first documented by Ouyang et al. (2022) and Bai et al. (2022), who empirically demonstrated that the in-context learning capabilities of LLMs degrade after Reinforcement Learning from Human Feedback (RLHF). Subsequently, Kirk et al. (2023) revealed that RLHF fine-tuning also leads to a reduction in output diversity. These findings highlight the issue of overfitting during the fine-tuning process. Importantly, we emphasize that these limitations are not exclusive to RLHF but also manifest in SFT alone, as evidenced by (Ouyang et al., 2022, Figure 29) and O'Mahony et al. (2024). Given that SFT is the initial step in the post-training pipeline, addressing these limitations at their source is critical to improving the overall robustness and diversity of fine-tuned models.

Our argument—that CE is not the best fit for SFT—aligns with the findings of a recent study (Xiao, 2024). This research highlights that pre-training with CE operates within a "scaling-centric" regime, where the generalization loss aligns closely with the training loss, rendering the generalization gap negligible and making CE a viable choice. However, in the context of SFT, CE loss falls into the "generalization-centric" regime, where careful consideration of regularization becomes crucial.

**Game-Theoretic Formulation in LLMs.** Closely related to our work, recent studies (Chen et al., 2024; Li et al., 2024a) explored improving CE-trained models using techniques like self-play from game theory. However, our approach differs in two key ways. First, we introduce the maximum

entropy principle into distribution matching, while their work examines the standard distribution matching framework. Second, we focus on addressing the limitations of CE loss by designing methods that directly applies to pre-trained models, whereas their methods are applied post-SFT.

**Imitation Learning and Inverse Reinforcement Learning.** Our work is closely connected to Imitation Learning (IL) (Argall et al., 2009; Osa et al., 2018), a framework in which a learner acquires decision-making strategies by mimicking expert demonstrations. In fact, SFT can be interpreted as a form of IL with deterministic transitions (Sun & van der Schaar, 2024; Li et al., 2024a). Specifically, the cross-entropy loss used in SFT corresponds to behavior cloning (BC) in IL (Pomerleau, 1991), a method that directly replicates expert actions. However, our framework is more closely aligned with Generative Adversarial Imitation Learning (GAIL) (Ho & Ermon, 2016), which has been shown to generally outperform behavior cloning (Ke et al., 2019; Xu et al., 2020) by leveraging additional transition information to better match the expert policy distribution.

Our work shares conceptual similarities with maximum entropy Inverse Reinforcement Learning (IRL) (Ziebart et al., 2008), which introduces an entropy term to enhance the robustness of reward function estimation. However, a key distinction lies in objectives: while maximum entropy IRL aims to recover a reward function (under which the optimal policy is typically deterministic), our goal is to learn a generative distribution that captures diversity, rather than focusing solely on reward inference.

**Entropy Regularization.** Dubey et al. (2018) proposed that achieving zero CE loss is not essential for high accuracy. Instead, they suggested that a conditional probability distribution where the argmax corresponds to the correct class is sufficient. This concept motivates our use of entropy regularization, which allows for assigning probabilities to alternative options beyond the observed data. Prior to our work, Pereyra et al. (2017) also explored entropy regularization in the context of neural network training. It is important to note that Pereyra et al. (2017) focused on image classification tasks, while our focus is on text generation where data is sequential in nature and is more challenging. In the context of LLMs, Hu et al. (2023) explored the maximum entropy regularization by using GFlowNet (Bengio et al., 2021), but their methods require a reward function rather than supervised data.

## A.1 DIFFERENCE WITH SPIN

Closely related to our work, Chen et al. (2024) also presented a game formulation for fine-tuning LLMs and developed an algorithm called SPIN. However, their work differs fundamentally from ours. We highlight the key differences to help readers better understand the distinctions between our work and theirs:

- Motivation: SPIN aims to further leverage supervised data after optimizing the model with the CE loss, as Chen et al. (2024) believe SFT does not fully exploit the supervised data. In contrast, we focus on the limitations of SFT, particularly poor diversity and knowledge forgetting, and propose preserving output diversity to address these issues.

- Formulation: Chen et al. (2024) develop a theory proving that the equilibrium point converges to the distribution matching problem of the data distribution, which is reasonable at the population level. However, in practice, data distributions usually have limited size and coverage. To address this, we propose incorporating entropy regularization into the distribution matching framework.

- Technical approach: SPIN is inspired by techniques from DPO (Rafailov et al., 2023). Specifically, it introduces KL regularization in the algorithm design (rather than in the distribution matching formulation). As a result, SPIN is conceptually similar to the online version of DPO, where supervised data is treated as the chosen data and model-generated data as the rejected data. In contrast, entropy regularization is introduced directly in our game formulation, rather than at the algorithm design level. Furthermore, our approach is built on the proposed probability transfer theory, and our algorithm leverages an auxiliary variable that governs the distribution matching process to preserve output diversity. In essence, the two algorithms exhibit entirely different optimization properties and dynamics.

- Computational Efficiency: Our approach runs at the same speed and memory requirements as CE, while SPIN requires more computational resources due to its need for online sampling and the introduction of a reference model in its KL-regularized optimization. As a result, SPIN runs slower than our method.

## B  IMPLEMENTATION OF GEM

**Generalized GEM.** We note that we can generalize the formulation in Equations (3) and (4) by introducing a transformation function $h$ on $f$'s objective:

$$\min_f \quad \mathcal{L}_{\text{GEM}}(f, q) \triangleq \mathbb{E}_x \mathbb{E}_{y^{\text{real}} \sim p(\cdot|x)} \mathbb{E}_{y^{\text{gene}} \sim q(\cdot|x)} h\left(\left[\log f(y^{\text{gene}}|x) - \log f(y^{\text{real}}|x)\right]\right) \quad (9)$$

$$\max_q \quad \mathcal{Q}(f, q) \triangleq \mathbb{E}_x \mathbb{E}_{y^{\text{gene}} \sim q(\cdot|x)} \left[\log f(y^{\text{gene}}|x)\right] + \beta \cdot \mathcal{H}(q(\cdot|x)). \quad (10)$$

In this way, we have

$$-\nabla_\theta \mathcal{L}_{\text{GEM}}(f_\theta, q) = \sum_{j=1, j \neq i}^{K} w_{i \leftarrow j} \cdot e_{i \leftarrow j}, \quad (11)$$

$$w_{i \leftarrow j} = q(j|x) h'(\log f(y^{\text{gene}}|x) - \log f(y^{\text{real}}|x)).$$

The choice of the function $h$ directly affects the magnitude of $w_{ij}$. If $h(z) = z$ (i.e., a linear function presented in the main text), we simply have $h'(z) \equiv 1$, resulting in uniform weighting for probability adjustments. This corresponds to the algorithmic design in Section 4. Inspired by (Jolicoeur-Martineau, 2019; Sun et al., 2020; Rafailov et al., 2023), we also explore the design $h(z) = \log \text{sigmoid}(-z)$, then $h'(z) = \text{sigmoid}(z)$. In this case, $w_{i \leftarrow j}$ becomes adaptive: it takes a higher value when $\log f(y^{\text{gene}}|x) > \log f(y^{\text{real}}|x)$ and a lower value otherwise. This adaptability ensures that the model focuses more on cases where it struggles to distinguish between real and generated data.

In our experiments, we actually implement the $\log \text{sigmoid}$ function for $h$, as it provides more adaptivity than the linear function. We note that the linear function also works well in practice, but the $\log \text{sigmoid}$ function yields slightly better performance in some tasks.

**Extension to Sequential Data.** In the main text, we have derived the algorithm for the case $y$ is non-sequential. We note that optimization in the sequential case could be highly difficult. With a little abuse of notations, let $y = (y_1, \ldots, y_T) \triangleq y_{1:T}$. Now, we can extend the formulation in Equation (3) to the following:

$$\min_f \quad \mathbb{E}_x \mathbb{E}_{y_{1:T}^{\text{real}} \sim p(\cdot|x)} \mathbb{E}_{y_{1:T}^{\text{gene}} \sim q(\cdot|x)} \left[h\left(\log \log f(y_{1:T}^{\text{gene}}|x) - f(y_{1:T}^{\text{real}}|x)\right)\right] \quad (12)$$

$$\max_q \quad \mathbb{E}_x \mathbb{E}_{y_{1:T}^{\text{gene}} \sim \pi(\cdot|x)} \left[\log f(y_{1:T}^{\text{gene}}|x)\right] + \beta \cdot \mathcal{H}(\pi(\cdot|x))$$

Here, we encounter a challenge: the joint distribution of $y_{1:T}$, as a cascaded categorical distribution, is quite complicated. This results in the expectation $\mathbb{E}_{y_{1:T}^{\text{gene}}}[\cdot]$ cannot be easily calculated as before.

To deal with the above challenges, we propose decomposing the multi-step sequential optimization problem into multiple single-step optimization problems and solve each efficiently. This is inspired by the data distribution "reset" trick introduced by (Ross et al., 2011) in imitation learning, where the teacher first demonstrates a few actions, and the student completes the reset. For our problem, we restrict the distribution matching to the case that the prefix samples up to time step $t$ are drawn from the data distribution $p$ and solves the optimization problem at the $t$-th time step as before. Its mathematical formulation is given below:

$$\max_f \mathcal{L}^{\text{seq}}(f, q) = \mathbb{E}_x \left\{ \sum_{t=1}^{T} \mathbb{E}_{y_{1:t-1}^{\text{real}} \sim p(\cdot|x)} \mathbb{E}_{y_t^{\text{real}} \sim p(\cdot|x, y_{1:t-1}^{\text{real}})} \mathbb{E}_{y_t^{\text{gene}} \sim q(\cdot|x, y_{1:t-1}^{\text{real}})} [\Delta] \right\} \quad (13)$$

$$\text{where} \quad \Delta = \left[h\left(\log f(y_t^{\text{gene}}|x, y_{1:t-1}^{\text{real}}) - \log f(y_t^{\text{real}}|x, y_{1:t-1}^{\text{real}})\right)\right],$$

The main advantage of this formulation is that for each sub-problem, we still have access to the conditional distribution, allowing the previously discussed computational advantages to remain applicable. The same idea applies to the training of distribution $q$, so we still have the closed-form solution that $q(\cdot|x, y_{1:t-1}^{\text{real}}) = \text{softmax}(1/\beta \cdot \log f(\cdot|x, y_{1:t-1}^{\text{real}}))$.

We outline the proposed procedure for dealing with sequential data in Algorithm 2 and provide its PyTorch implementation below. We examine the importance of the "reset" technique in Appendix F.

---

**Algorithm 2** GEM for Sequential Data

---

**Input:** Dataset $\mathcal{D} = \{(x_i, y_1, \ldots, y_T)\}$
1: Initialize $\widetilde{\mathcal{D}} = \emptyset$
2: **for** sample index $i$ **do**             ▷ *"Reset" data distribution*
3:      **for** timestep index $t = 1, \ldots, T$ **do**
$$\widetilde{x} = x_i \oplus (y_1^{\text{real}}, \ldots y_{t-1}^{\text{real}}), \quad \widetilde{y} = y_t^{\text{real}}$$
$$\widetilde{\mathcal{D}} \leftarrow \widetilde{\mathcal{D}} \cup \{(\widetilde{x}, \widetilde{y})\}$$
4: $f_\theta \leftarrow$ Call **Algorithm 1** on $\widetilde{\mathcal{D}}$
**Output:** Generative model $f_\theta$

---

```python
def gem_loss(logits, labels, beta=0.7, ignore_index=-100, h="linear"):

    shift_logits = logits[..., :-1, :].contiguous()
    shift_labels = labels[..., 1:].contiguous()

    mask = shift_labels != ignore_index
    shift_logits = shift_logits[mask]
    shift_labels = shift_labels[mask]

    with torch.no_grad():
        logits_on_labels = torch.gather(
            shift_logits, dim=-1, index=shift_labels.unsqueeze(-1)
        ).squeeze(-1)

        logits_diff = shift_logits - logits_on_labels.unsqueeze(-1)
        if h == "linear":
            weights = torch.ones_like(logits_diff)
        elif h == "log_sigmoid":
            weights = F.sigmoid(0.01 * logits_diff)
        else:
            raise ValueError(h)

    gene_log_probs = F.log_softmax(shift_logits, dim=-1)
    q_probs = torch.exp(
        F.log_softmax(shift_logits / beta, dim=-1)
    ).detach()

    real_log_probs = torch.gather(
        gene_log_probs, dim=-1, index=shift_labels.unsqueeze(-1)
    ).squeeze(-1)

    loss = -torch.sum(
        q_probs * weights * (real_log_probs.unsqueeze(-1) -
    gene_log_probs), dim=-1
    ).mean()

    return loss
```

Listing 1: Pytorch Code of GEM

To understand the above implementation, we note that we leverage the gradient analysis in Section 5.3: first, we calculate the re-weighting term in Lines 9–20. Then, we calculate the difference in log-probabilities in Lines 22–33. Note that we use a coefficient of $0.01$ to scale the input in the `log-sigmoid` function. This ensures that the function behaves nearly linearly.

**Computational Complexity Analysis**: We observe that the computational complexity of GEM is nearly equivalent to that of optimizing CE loss. To clarify, the computational cost of CE involves two primary steps: a forward pass through the Transformer to compute the distribution $f_\theta$, followed by calculating the likelihood $\log f_\theta(y|x)$. The forward pass, which entails multiple layers of matrix multiplications, is the primary computational bottleneck.

Similarly, GEM requires one forward pass through the Transformer to compute the distributions $q$ and $f$, and then calculate the relative difference $\log f_\theta(y^{\text{real}}|x) - \log f_\theta(y^{\text{gene}}|x)$. As with CE, the forward pass is the main bottleneck in GEM. Backpropagation is performed in a comparable manner for both methods, resulting in GEM achieving nearly the same training speed as CE.

In terms of memory consumption, GEM requires storing an additional distribution $q$, which occupies the same amount of memory as $f$. For example, in our setup with a batch size of 4, a sequence length of 2048, and a vocabulary size of 128k, $q$ requires only about 2 GB of memory. This is negligible compared with the memory consumed by other training components such as gradients, optimizer states, and activation caches, which can collectively exceed 100 GB.

## C    PROOF

**Proposition 3.** *For the entropy-regularized KL minimization problem in Equation* (8)*, in the function space, we have the optimal solution:*

$$f^\star(y|x) = \frac{1}{Z_x} p(y|x)^{1/(\gamma+1)}$$

*where $Z_x$ is a normalization constant $\sum_{y'} p(y'|x)^{1/(\gamma+1)}$.*

The proof is based on the optimality condition of constrained optimization. Its proof can be found in the previous literature (see, e.g., (Vieillard et al., 2020, Appendix A)). We note that the above closed-form solution cannot be applied in practice because we do not have access to the density function of the data distribution $p$.

*Proof of Proposition 2.* **Proof of the first part.** When $h$ is a linear function, we have that

$$\mathcal{L}_q(f)$$
$$= \mathbb{E}_x \mathbb{E}_{y^{\text{real}} \sim p(\cdot|x)} \mathbb{E}_{y^{\text{gene}} \sim q(\cdot|x)} \left[ \log f(y^{\text{real}}|x) - \log f(y^{\text{gene}}|x) \right]$$
$$= \mathbb{E}_x \mathbb{E}_{y^{\text{real}} \sim p(\cdot|x)} \mathbb{E}_{y^{\text{gene}} \sim q(\cdot|x)} \left[ \log f(y^{\text{real}}|x) \right] - \mathbb{E}_x \mathbb{E}_{y^{\text{real}} \sim p(\cdot|x)} \mathbb{E}_{y^{\text{gene}} \sim q(\cdot|x)} \left[ \log f(y^{\text{gene}}|x) \right]$$
$$= \mathbb{E}_x \mathbb{E}_{y^{\text{real}} \sim p(\cdot|x)} \left[ \log f(y^{\text{real}}|x) \right] - \mathbb{E}_x \mathbb{E}_{y^{\text{gene}} \sim q(\cdot|x)} \left[ \log f(y^{\text{gene}}|x) \right]$$

For any $x \in \mathcal{X}$, we have that

$$\frac{\partial \mathcal{L}}{\partial f} = \frac{p - q}{f} \tag{14}$$

When $q = p$, we have $\partial \mathcal{L}/\partial f = 0$, i.e., a stationary point. By examining Equation (7), we have that $f^\star = \texttt{softmax}(\beta * \log p) = \texttt{softmax}(\beta * \log q^\star)$, so $q^\star$ is also a stationary point. This proves that the point in Equation (7) is a stationary point for the Nash game.

To prove that it is the unique stationary point, we use a contraction-based argument. Assume there exists another equilibrium point in addition to the one in Equation (7). This implies that either $q^\star \neq p$ or $f^\star \neq \texttt{softmax}(1/\beta \log f)$. In the former case, the gradient in Equation (14) cannot be zero, which leads to a contradiction. In the latter case, $q^\star$ cannot be the optimal solution to Equation (6), again resulting in a contradiction. Thus, the original claim regarding uniqueness holds.

**Proof of the second part.** It is based on Proposition 3. As analyzed in Proposition 3, for $\beta = 1/(\gamma + 1)$, the optimal $f$ in Equation (7) corresponds to the the optimal solution of minimizing reverse KL with entropy regularization.

$\square$

## D    DISCUSSION

### D.1    TEMPERATURE ADJUSTMENT FOR OUTPUT DIVERSITY

Temperature adjustment is a widely used trick for encouraging output diversity. However, this trick is applied during the **inference** stage, which differs from our focus on preserving output diversity during **training**. Moreover, the mechanisms for increasing diversity are fundamentally different.

When the temperature of the softmax distribution is increased, the probability mass shifts away from the mode (the most likely outcomes) toward the tail (less likely outcomes). While this can enhance diversity, it also carries a significant drawback: in LLMs, the tail of the distribution often contains many nonsensical or low-quality tokens. As a result, increasing the tail probability can lead to undesirable or incoherent outputs.

In contrast, GEM employs a different mechanism to protect diversity. By promoting sparse updates and preserving tail probabilities, GEM ensures that diversity is maintained without disproportionately amplifying nonsensical or low-quality tokens. This approach provides a more controlled and effective way to balance diversity and output quality during training.

## D.2 THEORY OF CE

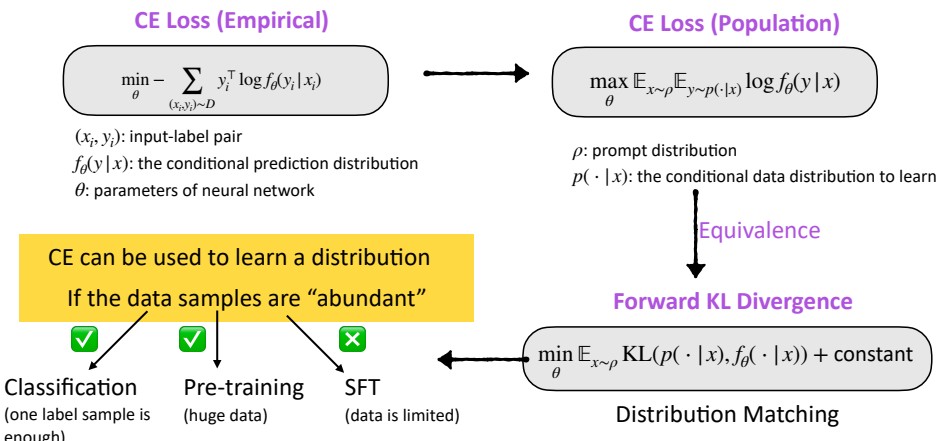

Figure 10: Theory of CE. The figure illustrates the connection between the empirical (top left) and population (top right) versions of CE, as well as its relationship to distribution matching (bottom right). It also highlights the applicable regime of CE (bottom left).

CE is a fundamental loss function in machine learning that serves as a method for learning probability distributions. As illustrated in Figure 10, CE has both empirical and population formulations, where the empirical version minimizes the negative log-likelihood of predictions given inputs across training data, while the population version maximizes the expected log-likelihood across the true data distribution. In particular, its population formulation is equivalent to minimizing the KL divergence between the true conditional distribution and the model's predicted distribution (plus a constant), making CE essentially a distribution matching technique. CE's effectiveness depends critically on the quantity of data available to approximate the underlying distribution.

As discussed in our introduction, CE performs well for classification tasks where even a single labeled example per class can provide sufficient signal, and for pre-training scenarios with abundant data. However, it typically underperforms for SFT of LLMs where data is limited and generation tasks are inherently open-ended.

## D.3 CE WITH WEIGHT DECAY

Weight decay is a widely used technique for mitigating overfitting, particularly effective in training convolutional neural networks (CNNs). However, we observed that it is not always effective when applied to training generative models with Transformers. We hypothesize two primary reasons for this discrepancy:

- **Architecture.** The structural characteristics of CNNs make them more homogeneous in design, resulting in a relatively uniform loss landscape across different parameter blocks within the network.

In contrast, Transformers exhibit heterogeneous properties, as noted in recent works (Zhang et al., 2024; Ormaniec et al., 2024), leading to significantly varied loss landscapes across parameter blocks. As a result, a uniform weight decay applied to all parameters in Transformers may be suboptimal, since different parameter blocks might require specialized weight decay strategies.

- **Task**. CNNs in classification tasks are designed to learn a predictor that outputs a unique prediction, whereas generative models aim to learn a distribution. For generative models, directly regularizing the distribution is preferable. Weight decay, which regularizes parameters indirectly, may not effectively serve this purpose.

Based on these observations, we argue that entropy regularization is better suited for training generative models with Transformers. Unlike weight decay, entropy regularization directly focuses on the target distribution, and its influence can effectively backpropagate to specific parameters through adaptive optimizers like Adam, accommodating the heterogeneity of Transformer architectures. A more deeper exploration of this topic is left for future work.

### D.4 CE WITH ENTROPY REGULARIZER

Inspired by our framework, one may propose an alternative approach for preserving output diversity is to introduce an entropy regularization with the CE loss. We discuss this approach in this section.

Formally, we have that

$$\max_f \mathbb{E}_x \Big\{ \underbrace{\mathbb{E}_{y \sim p(\cdot|x)}[\log f(y|x)]}_{=-D_{\mathrm{KL}}(p,f)+\text{constant}} + \gamma \cdot \underbrace{\mathbb{E}_{y \sim f(\cdot|x)}[-\log f(y|x)]}_{=\mathcal{H}(f)} \Big\} \tag{15}$$

This corresponds to the forward KL with maximum entropy regularization. This corresponds to the forward KL divergence with maximum entropy regularization. We note that this optimization problem does not have a closed-form solution. To better understand its behavior, we observe that this formulation leads to an improper increase in tail probabilities when maximizing the entropy, as illustrated in Figure 11. In the context of LLMs, this increase often translates into nonsensical tokens in the vocabulary, leading to undesirable generation outputs. A concrete example is provided in Table 1, in which we directly sample sentences.

Table 1: Examples of generated sentences for models trained with the `UltraFeedback` dataset. The symbol "[...]" indicates good sentences that have been omitted to save space. This example shows that CE with entropy regularization can generate nonsensical tokens (shown in red) due to the increase in tail probabilities.

| Prompt | Whats the largest star in our galaxy? |
|---|---|
| CE | Hello! Atlantis is a legendary city that was said to have existed in ancient Greece. According to the story, it was a highly advanced and prosperous city that was located on an island in the ocean. [...] |
| CE+Entropy | Hello! Atlantis Documentary is a 2019 American documentaryéhoFLICT film directed by Já oblík and produced by Werner Herzog. The film explores the history and legacy of Atlantis, □ an ancient Greek city-state that was said to have_calendar knowledge and advanced technology, through interviews with scholars and historians.ython |
| GEM-LS | Hello there! As a helpful, respectful, and honest assistant, I'd be happy to help you explore the fascinating topic of Atlantis! Atlantis is an ancient Greek myth that tells the story of a legendary realm said to have existed in the Atlantic Ocean, west of the Pillars of Hercules. [...] |

The core issue of CE with an entropy regularizer arises because **the gradient of the entropy regularizer can dominate for tokens with low probabilities**. Specifically, we have that

$$\frac{\partial - D_{\mathrm{KL}}(p,f)}{\partial f} = -\frac{p}{f}, \quad \frac{\partial \mathcal{H}}{\partial f} = -(1 + \log f).$$

where the division is element-wise. Consequently, for tokens with low probabilities in both $f$ and $p$, i.e., $f(x) \approx 0$ and $p(x) \approx 0$, the gradient from the forward KL divergence, $-p(x)/f(x)$, could be of

constant order. In contrast, the gradient from the entropy regularizer, $(-1 + \log f(x)) \to -\infty$ as $f(x) \to 0$. This imbalance disproportionately increases the tail probabilities, leading to undesirable effects.

In contrast, the proposed method, GEM, does not have this issue. This is because both distribution matching and entropy regularization in GEM are defined jointly over the generative distribution $f$, rather than being applied separately to the data distribution $p$ and generative distribution $f$. As shown in the proof of Proposition 2, the gradient of GEM is given by:

$$\frac{\partial - \mathcal{L}}{\partial f} = \frac{-p + q}{f} = -\frac{p}{f} + \frac{q}{f},$$

where the first term, $-p/f$, is identical to that of CE with entropy regularization, but the second term $q/f$, is unique to GEM. Since $q = \texttt{softmax}(1/\beta * \log f)$, $q$ is a more squeezed distribution than $f$. Consequently, for $f(x) \approx 0$, we have $q(x)/f(x) < 1$. This ensures that the gradient is not dominated in the low-probability (tail) region, preventing improper increases in tail probabilities. Thus, GEM achieves a more balanced optimization.

### D.5 RELATION WITH GAN

We clarify that while our framework shares a similar structure with GANs (Goodfellow et al., 2014; Jolicoeur-Martineau, 2019), it has a totally different meaning. Specifically, GANs were originally designed for image generation tasks, where a discriminator network is additionally introduced to measure the distance between distributions of real and generated images. We do not follow this storyline. Unlike in GANs, where measuring distances between image distributions is challenging, computing distances between token distributions in language models is simple due to the discrete nature of token distributions. As we have explained, the introduction of the variable $q$ in our framework is to control the direction and magnitude of probability transfer during distribution matching—an objective that is distinct from the goal of GANs.

We note that Proposition 3 also has an important computational implication. While the distribution matching problem in Equation (8) is theoretically well-defined, it is computationally intractable in practice. In contrast, our game-theoretic formulation provides a feasible alternative. To understand why, consider the decomposition of the reverse KL divergence:

$$D_{\mathrm{KL}}(f(\cdot|x), p(\cdot|x)) = \sum_y f(y|x) \log \frac{f(y|x)}{p(y|x)} = \sum_y f(y|x) \log f(y|x) - \sum_y f(y|x) \log p(y|x).$$

In practice, we only have access to finite samples drawn from $p$ rather than direct knowledge of $\log p(y|x)$. Consequently, the term $\sum_y f(y|x) \log p(y|x)$ cannot be easily estimated from finite samples, rendering the formulation in Equation (8) impractical. This difficulty contrasts with the CE formulation, which corresponds to a forward KL divergence that can be easily estimated from finite samples. We note that a similar situation arises in GANs (Goodfellow et al., 2014), where the game-theoretic approach effectively solves the Jensen-Shannon divergence minimization problem, even though the Jensen-Shannon divergence itself cannot be directly estimated from data samples.

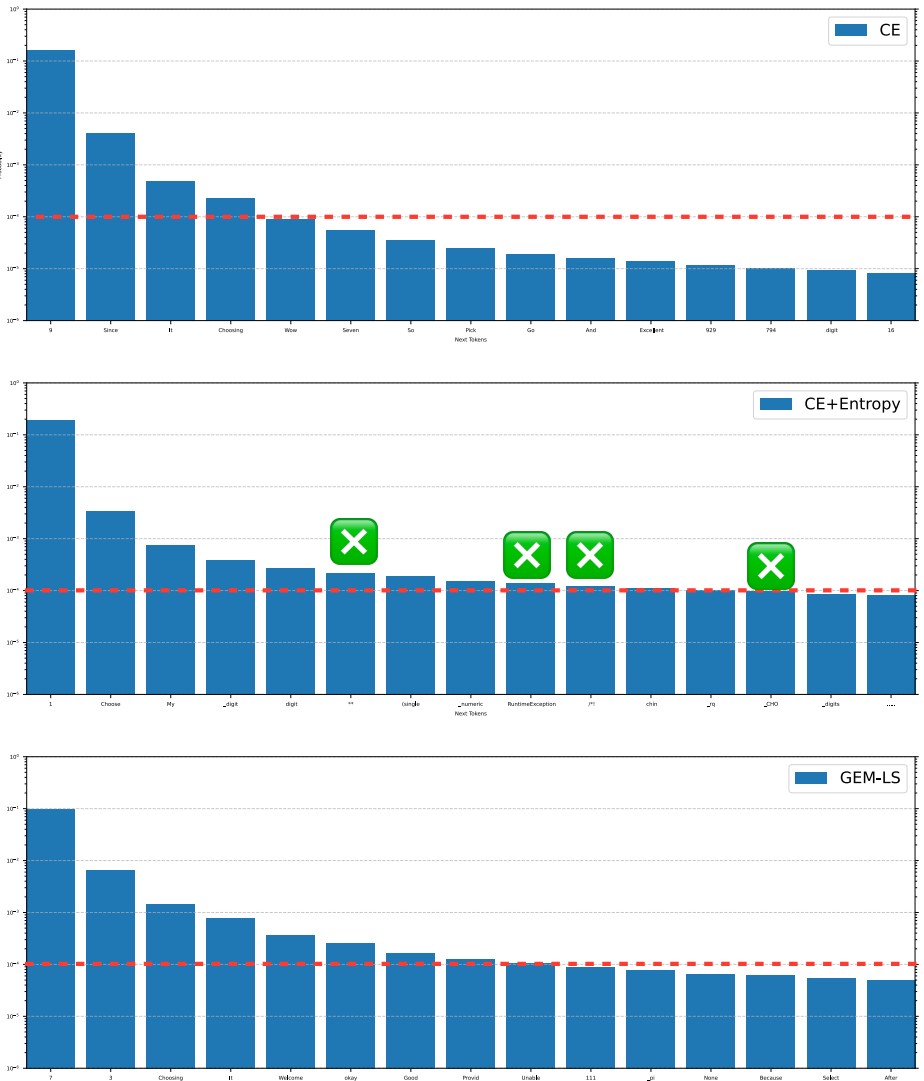

Figure 11: Distributions of next-token probabilities for trained models with the `UltraFeedback` dataset, presented from top to bottom: CE, CE+Entropy, GEM-LS. The prompt is "Give me a single-digit number". The top 300 probabilities are shown with a subsampling rate of 20 for clear visualization. A red dotted line indicates the probability threshold of $10^{-4}$. The figure demonstrates that the CE+Entropy model has a longer tail with higher probabilities assigned to some nonsensical tokens, marked with crosses.

# E  EXPERIMENT DETAILS

All experiments are conducted using A800-80GB GPUs with the DeepSpeed distributed training framework, utilizing ZeRO-2 and gradient checkpointing without offloading. We use flash-attention-2 with deterministic backward for reproducibility. The experiments are based on the pretrained Llama-3-8B model, using Adam as the optimizer with a global batch size of 128. Following (Yu et al., 2023; Liu et al., 2023; Cui et al., 2024), the learning rate is set to $2 \times 10^{-5}$, with a warm-up ratio of 0.03 and cosine learning rate decay. Training is performed over 3 epochs. All supervised datasets are formatted into the chat format using the Llama-3.1-8B-Instruct's tokenizer. When generation of responses is required for evaluation, we use the vLLM to accelerate inference.

## E.1  ULTRAFEEDBACK

We use the dataset filtered by HuggingfaceH4 team, which is available at `https://huggingface.co/datasets/HuggingFaceH4/ultrafeedback_binarized`. The dataset contains 61,135 training samples and 1,000 test samples. For training, we set the maximum sequence length to 2,048, dropping longer sequences and padding shorter ones. To achieve a global batch size of 128, we use a per-device batch size of 4, a gradient accumulation step of 4, and 4 GPUs. The training times takes about 24 GPU hours for all methods. For the CE method, we have tuned hyperparameters for weight decay and entropy regularization, selecting values from $\{0.1, 0.01, 0.001\}$. In both cases, a value of 0.1 provided the best overall results. For NEFT, we use a noise scale hyperparameter of 5, as recommended by (Jain et al., 2023).

Evaluation metrics, including perplexity, and entropy, are based on these 1,000 test samples. For entropy calculation, we compute the conditional entropy, whose expectation can be calculated exactly, and average over the sequence.

For the chatting task, we use the 805 test questions from the `AlpacaEval` dataset and employ the reward model `FsfairX-LLaMA3-RM-v0.1`. The maximum generation sequence length is set to 2048. For each question, 32 samples are generated with the configuration `temperature=0.6`, `top_k=50, top_p=0.9`. To calculate the win rate, we use the Bradley-Terry model:

$$\mathbb{P}(y \succ y' \mid x) = \frac{\exp(r(x, y))}{\exp(r(x, y)) + \exp(r(x, y'))}.$$

We use GPT-4 generated responses as a baseline for calculating the win rate, specifically the `gpt4_1106_preview`[8] version.

For the code generation task, there are 164 test questions for `HumanEval`. We use the prompt from (Wei et al., 2024):

> You are an exceptionally intelligent coding assistant that consistently delivers accurate and reliable responses to user instructions.
> @@ Instruction
> {instruction}

For each question, 200 responses are generated with the configuration `temperature=0.6`, `top_k=50, top_p=0.9` to estimate the pass rate. The evaluation scripts are from `https://github.com/ise-uiuc/magicoder/blob/main/experiments/text2code.py`.

# F  ADDITIONAL RESULTS

## F.1  DIVERSITY REDUCTION OF SFT VIA CE

We provide empirical evidence that while SFT using CE loss improves the instruction-following ability of models for downstream tasks, it significantly reduces the output diversity of pre-trained models. To demonstrate this, we conduct experiments using the pre-trained LLM, Llama-3.1-8B,

---

[8]`https://github.com/tatsu-lab/alpaca_eval/blob/main/results/gpt4_1106_preview/model_outputs.json`

employing few-shot learning to teach the model to follow instructions and answer questions. We compare the output diversity of the pre-trained LLM with that of the same model fine-tuned using CE loss. Specifically, we prompt both models to answer 100 questions from the AlpacaEval dataset (Li et al., 2023), generating 32 responses for each question. Following (Kirk et al., 2023), we then measure output diversity using the following metrics: 1) N-gram diversity: the proportion of distinct n-grams in a single response (intra-diversity); 2) Self-BLEU diversity: calculated as 100 minus the Self-BLEU score (inter-diversity), where one response is treated as a reference among multiple generated responses; 3) Sentence-BERT diversity: the cosine dissimilarity between pairs of responses in the embedding space. All criteria range from 0 to 100 (with Sentence-BERT diversity scaled by multiplying by 100), and higher values indicate greater diversity.

The results, presented in Table 2, show that fine-tuning with CE loss reduces output diversity compared to the pre-trained LLM. For completeness, we also report the output diversity of GEM in this setting. GEM demonstrates better output diversity than the CE-tuned model and performs comparably to the pre-trained LLM—in fact, it shows a slight improvement. We believe this marginal improvement is partially attributable to the few-shot prompting strategy, which slightly restricts the output space of the pre-trained LLM.

Table 2: Output diversity of pre-trained, CE fine-tuned, and GEM fine-tuned LLMs. Higher values indicate greater diversity.

|  | N-gram Diversity | Self-BLEU Diversity | Sentence-BERT Diversity |
|---|---|---|---|
| Pre-trained LLM (5-shot) | 23.3 | 73.6 | 14.2 |
| CE | 22.5 | 71.1 | 13.9 |
| GEM | 25.8 | 75.4 | 15.4 |

## F.2 SENSITIVITY OF HYPER-PARAMETERS

In this main text, our theory suggests that the hyper-parameter $\beta$ controls the strength of entropy regularization, with $\beta \to 0$ corresponding to no regulation and $\beta \to 1$ corresponding to strong entropy regularization. We believe such guidance is sufficient for practitioners. Additionally, we provide sensitivity studies on the chat task in Table 3. We observe that as $\beta \to 1$, the performance of GEM converges to that of CE, and as $\beta \to 0$, GEM benefits from diversity preservation.

Table 3: Performance on `Chat` task of Llama-3.1-8B when trained with the `UltraFeedback` dataset.

|  | Random Sampling | BON@32 |
|---|---|---|
| CE | 24.5 | 47.3 |
| GEM ($\beta = 1.0$) | 24.7 | 47.5 |
| GEM ($\beta = 0.9$) | 24.7 | 48.3 |
| GEM ($\beta = 0.8$) | 24.8 | 49.4 |
| GEM ($\beta = 0.7$) | 24.9 | 50.4 |
| GEM ($\beta = 0.6$) | 24.3 | 50.1 |

## F.3 MATH REASONING TASKS

In this section, we demonstrate that GEM is also effective in mathematical reasoning tasks. We evaluate the model trained with the `UltraFeedback` dataset on the GSM8K (Cobbe et al., 2021) task. We use the following prompt to generate responses:

> Your task is to answer the question below. Give step-by-step reasoning before you answer, and when you're ready to answer, please use the format "The answer is: ...".
> Question: {question}

We consider three metrics: greedy decoding, majority voting (Wang et al., 2023), and pass rate. For the latter two metrics, we report the performance across 16 samples with a temperature of 0.6 and

top-p of $0.9$. The results are shown in Table 4. We find that GEM mitigates overfitting, as evidenced by improved performance in greedy decoding. Additionally, by preserving output diversity, GEM enhances performance in majority voting and pass rate.

Table 4: Performance on GSM8K (Cobbe et al., 2021) of Llama-3.1-8B when trained with the UltraFeedback dataset. For majority voting and pass rate, we use 16 samples.

|  | Greedy Decoding | Majority Voting | Pass Rate |
| --- | --- | --- | --- |
| CE | 49.7 | 67.3 | 88.8 |
| CE +WD | 50.0 | 65.9 | 89.5 |
| NEFT | 48.4 | 66.3 | 89.1 |
| GEM | 51.3 | 69.5 | 91.1 |

## F.4 SCALABILITY

In this section, we demonstrate the scalability of our method across models of varying sizes, ranging from 3B to 70B parameters, including Qwen-2.5-3B and Qwen-2.5-7B (Qwen, 2024), Gemma-2-9B (Gemma et al., 2024), and Llama-3.1-70B (Dubey et al., 2024). For all models except Llama-3.1-70B, we apply LoRA (Hu et al., 2022) with a rank of 16 due to computational constraints, while maintaining the same settings and hyperparameters used in our experiments with Llama-3.1-8B. The results presented in Figure 12 show that the benefits observed with Llama-3.1-8B extend to other models as well, confirming the broader effectiveness of GEM.

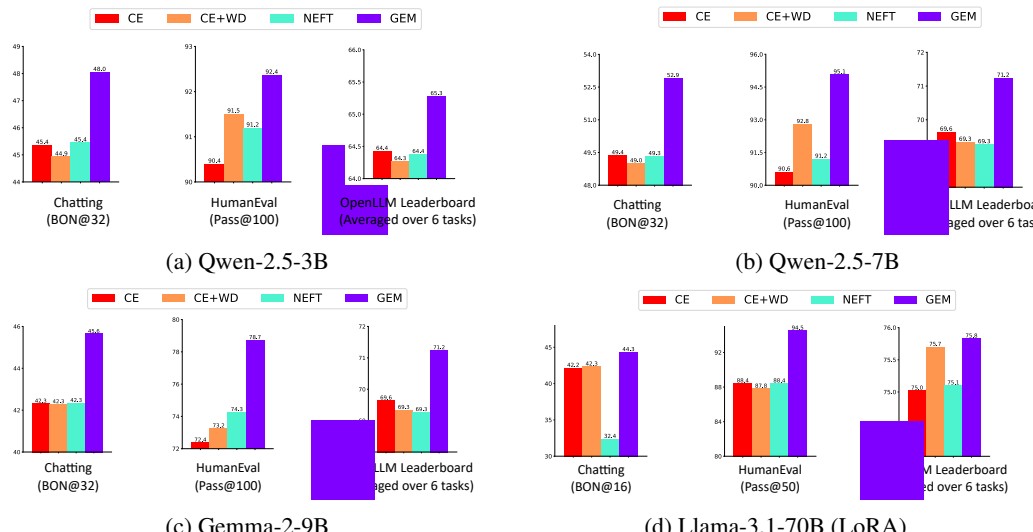

Figure 12: Performance across different architectures and model sizes. The results show that GEM consistently outperforms CE.

## F.5 IMPORTANCE OF DATA RESET

In this section, we extend the technical challenge in Section 5 regarding how to solve the problem in Equation (12). To recap, the problem is that

$$\max_f \quad \mathbb{E}_x \mathbb{E}_{y_{1:T}^{\text{real}} \sim p(\cdot|x)} \mathbb{E}_{y_{1:T}^{\text{gene}} \sim q(\cdot|x)} \left[ h \left( \log f(y_{1:T}^{\text{real}}|x) - \log f(y_{1:T}^{\text{gene}}|x) \right) \right]$$

$$\text{s.t.} \quad q = \operatorname*{argmax}_\pi \mathbb{E}_x \mathbb{E}_{y_{1:T} \sim \pi(\cdot|x)} \left[ \log f(y_{1:T}|x) \right] + 1/\beta \cdot \mathcal{H}(\pi(\cdot|x))$$

A key challenge is that the expectation $\mathbb{E}_{y_{1:T}^{\text{gene}}}[\cdot]$ cannot be calculated as easily as before. Worse still, Monte Carlo estimation as used in (Chen et al., 2024; Li et al., 2024a), by drawing samples from the distribution, does not provide an accurate gradient estimate. A fundamental difficulty in this stochastic approximation arises from the distribution shift between the SFT data and the pre-trained

distribution. To better understand this, refer to examples provided in Table 5. We observe that SFT data typically has finite-length sequences, while the pre-trained distribution produces samples that are repetitive and can even be infinite in length.

This causes issues: assuming the probability of any token is lower-bounded by a small number $c \in (0, 1)$, this means that $\log f(y_{1:T}^{\text{gene}}|x)$ approaches to $-\infty$ when $T$ goes to infinite for pre-training distribution data. While this might seem acceptable as the gradient would reduce the probability of such samples, the challenge is that the sample size is vast: for the Llama-3-8B model, the vocabulary size is 128k, and with a typical sequence length of 2048, the sample space size is $128000^{2048}$. This makes it difficult for the model to find effective directions for improvement. To validate this claim, we directly implemented the idea of stochastic approximation and found that training failed after 80 optimization steps (with 10k samples)[9], and the model could not generate good responses; see Table 5. In fact, techniques in (Chen et al., 2024; Li et al., 2024a) are usually applied to models after SFT, where the distribution shift between the model and data is smaller. This technical remark is also discussed in the online forum `https://github.com/uclaml/SPIN/issues/26#issuecomment-2062926716`.

## G  FUTURE WORK

Below, we explore GEM's potential applications.

**Better Cold-Start for Online RL Training.** SFT typically serves as a cold start for online RL training. To fully harness the potential of RL, exploration across diverse generated responses is critical. However, CE often reduces diversity, making it less effective in this context. Generalized Experience Replay (GEM) helps maintain diversity when learning from cold-start data, ultimately enhancing RL training performance. This idea is explored in a recent blog post https://www.notion.so/Can-Better-Cold-Start-Strategies-Improve-RL-Training-for-LLMs-17aa0742a51680828616c867ed53bc6b.

**Preference Collapse in RLHF.** SFT-trained models can be refined through Reinforcement Learning from Human Feedback (RLHF) to better align with human values (Ouyang et al., 2022; Bai et al., 2022). In this context, Xiao et al. (2024) studied the impact of SFT models on preference learning in RLHF, demonstrating that if an SFT model collapses (i.e., becomes biased toward certain outputs with near-certain probability), it can further lead to preference collapse in alignment. Their findings underscore the importance of addressing collapse during the SFT stage.

**Synthetic Data Generation.** SFT-trained models are often used as synthetic data generators for self-improvement (see, e.g., (Adler et al., 2024; Dubey et al., 2024)). In this context, maintaining output diversity is essential. By generating a wide range of diverse outputs, models can explore various potential solutions, reducing the risk of overfitting and uncovering better-performing strategies. Our experiments about best-of-n against a reward model Section 6 is inline of this topic.

**Mode Collapse.** When models are repeatedly fine-tuned on text generated by their predecessors, linguistic diversity gradually erodes. This recursive process amplifies existing errors and biases, with each successive generation inheriting the limitations of its predecessors. This phenomenon, known as mode collapse, has been extensively documented in prior studies (Guo et al., 2023; Shumailov et al., 2023). By introducing entropy regularizer, we expect to help mitigate mode collapse in the self-improvement.

---

[9]This method is computationally slow due to sampling responses. In fact, these computational resources of 80 steps are actually more than those required for GEM for 1 epoch tuning of 60k samples.

Table 5: Examples of generated sentences from pre-trained models. The symbol "[...]" indicates sentences have been omitted to save space for the ground truth label, and the repetitive sentences for the pretrained model. Examples illustrate the distribution shift between SFT data and the pre-trained distribution. In SFT data, ground truth labels have a finite length (marked by an explicit <EOS> in the example), whereas the pre-trained distribution often generates repetitive sentences without a clear endpoint.

| Prompt | I would like to know more about Clyde Hill Washington...what can you tell me? |
|---|---|
| Ground Truth Label | Clyde Hill is a small, affluent city located in King County, Washington, United States. It is situated on the eastern shore of Lake Washington, north of Bellevue, and east of Kirkland. It is known for its stunning views of the lake and the Cascade Mountains. |
| | Clyde Hill was incorporated in the year 1953 and is home to approximately 2,800 residents. The city has a mix of residential and commercial areas, with large estates and luxurious mansions, making it one of the wealthiest ZIP codes in the United States. The median household income is around $200,000, and the median home value is roughly $1.6 million. |
| | [...] |
| | Overall, Clyde Hill, Washington, offers a unique and luxurious lifestyle to its residents. The combination of picturesque scenery, luxury living, and excellent amenities makes it a truly special place. <EOS> |
| Pretrained Model | I would like to know more about Clyde Hill Washington...what can you tell me? |
| | I would like to know more about Clyde Hill Washington...what can you tell me? |
| | I would like to know more about Clyde Hill Washington...what can you tell me? |
| | I would like to know more about Clyde Hill Washington...what can you tell me? |
| | [...] |
| GEM without reset trick | Thank you for reaching out with your queries on Clyde Hill Washington! As of 2021, the estimated median home value in Clyde Hill, Washington, is $3,750,000. [...] Thank you for trusting me with your queries and I hope I was able to help. Stay in touch and talk to you soon! Hexatrigesimalerusform((␣␣))stdexcepts každ␣[%://);?¿¡/stdexcept |
| GEM | Clyde Hill is a small, affluent neighborhood located in the eastern part of Washington, D.C., bordering the Potomac River. It is known for its stunning views of the Washington Monument, the National Mall, and the Potomac River. The area is home to a mix of embassies, diplomatic missions, and luxury residential buildings. |
| | Clyde Hill is also home to the historic Clyde Hill House, which was built in 1929 and served as the official residence of the U.S. Ambassador to France from 1933 to 1946. The house is now a private residence and is not open to the public. |
| | [...] |
| | Overall, Clyde Hill is a unique and beautiful neighborhood that offers a glimpse into the history and elegance of Washington, D.C. <EOS> |

