# OpenReview forum: "Preserving Diversity in Supervised Fine-Tuning of Large Language Models"
_ICLR.cc/2025/Conference — ICLR 2025 Poster_

### Official Review · Reviewer_wHN6 · 2024-10-31

**Soundness:** 3
**Presentation:** 3
**Contribution:** 2
**Rating:** 5
**Confidence:** 4

**Summary:**

The paper introduces a novel method for supervised fine-tuning of large language models (LLMs) called Generative and Entropy-regularized Matching (GEM). GEM replaces the traditional Cross Entropy (CE) loss with a reverse Kullback-Leibler (KL) divergence minimization enhanced with entropy regularization. The approach seeks to mitigate overfitting and increase diversity in model outputs. The authors demonstrate that GEM outperforms CE in instruction-following, math reasoning, code generation, and creative writing tasks, using Llama-3-8B as a model. The method also claims computational efficiency by optimizing a single model and reducing sampling steps while preserving output quality.

**Strengths:**

* The paper addresses known limitations of CE in supervised fine-tuning, such as overfitting and limited diversity, which are crucial challenges for deploying LLMs in diverse tasks.
* GEM's performance is evaluated across multiple tasks and datasets, showing consistent improvements over CE in both general-purpose and domain-specific fine-tuning.

**Weaknesses:**

* While the authors attempt to adapt GEM for sequential data, the proposed solution (using a data distribution “reset” trick) might introduce limitations for real-time applications. This part could benefit from further empirical validation.
* The performance of GEM depends on parameters like the $\beta$ term in entropy regularization. The paper lacks a sensitivity analysis to show how robust GEM is to these hyperparameters
* Although GEM offers an alternative to Cross Entropy, the idea of using entropy regularization to promote diversity is not new, and reverse KL divergence has also been previously explored in generative modeling. The novelty in combining these may be seen as incremental, especially without substantial empirical differentiation from prior methods.

**Questions:**

* In scenarios with limited training data, does GEM perform consistently, or would CE still be preferable due to its simplicity?
* Would the regularization in GEM, especially the entropy component, introduce a non-trivial increase in compute or memory requirements, especially when scaling?
* The paper focuses on the Llama-3-8B model, but it is unclear if GEM’s performance gains generalize across various architectures or if they are specific to this model’s configuration.
* In cases where GEM aims to enhance diversity, could it inadvertently introduce biases in generation, particularly when generating responses with multiple valid answers?

---

> ### Author Response · Authors · 2024-11-21
> **Response (Part I)**
>
> Thank you for reading our paper and providing valuable feedback. We greatly appreciate your comments and have addressed your concerns and questions below.
>
> **Comment 1:** While the authors attempt to adapt GEM for sequential data, the proposed solution (using a data distribution “reset” trick) might introduce limitations for real-time applications. This part could benefit from further empirical validation.
>
> **Response 1:** Thank you for raising this concern. We assume you are referring to potential computational drawbacks (e.g., delays or increased computation time) introduced by the "reset" trick. If so, we would like to clarify that the "reset" trick does not introduce additional computational overhead: the training time for CE is 5h58m while the training time is 5h55m for GEM. However, if your concern lies elsewhere, we would be happy to address it further.
>
> We briefly explain the reason here. The "reset" trick does not introduce additional computational costs because the context prefix is directly available from the training data. While Line 3 and its associated for-loop are conceptually described in Algorithm 2, the implementation is much more efficient in practice. Specifically, the loss of GEM for different time steps can be computed in parallel using a single pass of the Transformer. This is equivalent to optimizing the standard CE loss. For further details, please refer to the PyTorch code provided in Appendix B, which demonstrates how the one-step distribution is efficiently computed across time steps in parallel. On the other hand, without using this reset trick, stochastic gradient estimation becomes time-consuming and ineffective, as we demonstrated in Appendix D.3.
>
> **Comment 2:** The paper lacks a sensitivity analysis to show how robust GEM is to these hyperparameters
>
> **Response 2**: Thank you for highlighting this concern. GEM introduces a single hyperparameter, $\beta$, and we have observed that its performance is robust to variations in this parameter. A sensitivity analysis is provided in Figure 8 of Appendix F.3, where $\beta$ was varied from 0.6 to 1.0. The figure is also available at the anonymous link: https://anonymous.4open.science/r/iclr2025_submission_5769-2E96/images/figure_8.png.
>
> The results demonstrate that GEM with other hyperparameter values also outperforms CE across multiple metrics, including instruction following, and mathematical reasoning, whether using greedy decoding or majority voting.
>
> **Comment 3:** Although GEM offers an alternative to Cross Entropy, the idea of using entropy regularization to promote diversity is not new, and reverse KL divergence has also been previously explored in generative modeling. The novelty in combining these may be seen as incremental, especially without substantial empirical differentiation from prior methods.
>
> **Response 3**: We appreciate the contributions of prior studies on reverse KL divergence and entropy regularization and have respectfully reviewed such works. However, we respectfully disagree with the characterization of GEM as merely an incremental combination of these ideas. Additionally, we argue that GEM demonstrates substantial empirical advantages over previous methods.
>
> - **Theoretical and Algorithmic Contributions**: While the ideas of reverse KL divergence minimization and entropy regularization have been explored in theory, their practical implementations poses significant challenges. Reverse KL minimization is notoriously difficult because the data density function is typically unknown, making a direct, tractable solution infeasible.  Although combining reverse KL minimization with entropy regularization appears promising in theory, practical implementations may depend on adversarial training techniques, similar to GANs. Unfortunately, such techniques have yet to demonstrate success at the scale of billion-parameter GPT models. In contrast, our work introduces a new and scalable training algorithm tailored for this problem, which we believe significantly advances the field and opens new avenues for advanced generative distribution matching.
> - **Empirical Superiority:** GEM shows *obvious* and *consistent* empirical advantages across many tasks, including instruction following, in-context learning, chatting, mathematical reasoning, and code generation. These advantages are demonstrated with less overfitting and better task-specific performance and output diversity. For instance, for challenging code generation benchmarks such as HumanEval and MBPP, GEM achieves a relative improvement of 10.7% over Cross Entropy (CE) and 5.6% over the best baseline method, as detailed in Section 5.1. These results provide strong evidence that GEM effectively reduces overfitting and delivers superior performance across diverse tasks.

---

> ### Author Response · Authors · 2024-11-21
> **Response (Part II)**
>
> **Question 4:**  In scenarios with limited training data, does GEM perform consistently, or would CE still be preferable due to its simplicity?
>
> **Response 4:** Thank you for highlighting this important consideration. Our empirical analysis indicates that GEM consistently outperforms CE in scenarios with limited data, including datasets as small as 5k samples. Detailed findings are provided in Appendix F.5. Below, we present a summary of key results from the GSM8K benchmark, where models were trained on varying amounts of the UltraFeedback dataset: 5k (8%), 10k (17%), 20k (33%), and 60k (100%).
>
> |                          | 5k   | 10k  | 20k  | 60k (all) |
> | ------------------------ | ---- | ---- | ---- | --------- |
> | CE (greedy decoding)     | 36.2 | 38.7 | 39.6 | 43.0      |
> | GEM-LS (greedy decoding) | **38.1** | **40.9** | **40.7** | **47.8**      |
> | CE (majority voting)     | 50.9 | 55.6 | 58.5 | 62.2      |
> | GEM-LS (majority voting) | **51.9** | **56.7** | **60.0** | **66.3**      |
>
> These results highlight that GEM delivers consistent improvements over CE in low-data regimes.
>
> **Question 5**: Would the regularization in GEM, especially the entropy component, introduce a non-trivial increase in compute or memory requirements, especially when scaling?
>
> **Response 5:** No, GEM is both scalable and as tractable as optimizing the CE loss. Its GPU memory consumption and training speed are equivalent to CE, as detailed in the complexity analysis in Appendix B. In practice, we have successfully applied GEM to train 70B-parameter models, achieving comparable computational efficiency while delivering superior performance to CE. These results are presented in Appendix F.5. Furthermore, we expect GEM to scale effectively for even larger models exceeding 100B parameters.
>
> **Question 6**: The paper focuses on the Llama-3-8B model, but it is unclear if GEM’s performance gains generalize across various architectures or if they are specific to this model’s configuration.
>
> **Response 6**: Thank you for raising this concern. We selected the Llama-3-8B model for our study because it was one of the most robust 8B-scale models available at the time of our research. To address your feedback, we extended our experiments to include models ranging from 3B to 70B, such as Qwen-2.5-3B, Qwen-2.5-7B, Gemma-2-9B, and Llama-3.1-70B. These models are state-of-the-art and were released in recent months.
>
> The results of these experiments are shown in Figure 11 in Appendix F, which is also accessible via the anonymous link: https://anonymous.4open.science/r/iclr2025_submission_5769-2E96/images/figure_11.png. Our findings indicate that baselines such as weight decay and NEFT often underperform relative to CE, while CE+Entropy produces results comparable to CE. Notably, GEM consistently demonstrates improvements over CE across all evaluated model sizes. These results provide strong evidence that GEM is not only well-suited for Llama-3-8B but also other models.
>
> **Question 7**: In cases where GEM aims to enhance diversity, could it inadvertently introduce biases in generation, particularly when generating responses with multiple valid answers?
>
> **Response 7**: Thank you for raising this insightful question. We have not identified any side effects related to bias. Specifically, our manual review of GEM-generated responses has not revealed any unintended biases beyond the expected diversity in outputs.
>
> Additionally, we evaluated the self-consistency of generated responses in the math reasoning task by calculating the majority vote ratio across responses. This ratio is approximately 53% across different methods, as outlined in our response to Question 4 of Reviewer xjk9. These findings indicate that GEM generates responses that are both diverse and consistent.
>
> If you have specific concerns or insights regarding potential biases, we would greatly appreciate your input and are eager to explore these potential limitations further.
>
>
> ----
>
> We sincerely thank you for your insightful review. We gratefully hope that you could re-evaluate our paper based on the revised manuscript and the clarifications provided above. If our responses have satisfactorily addressed your concerns, we would greatly appreciate it if you could consider updating your review score accordingly. However, if you have any additional concerns, please do not hesitate to let us know. We are more than willing to provide further clarification.

---

> > ### Author Response · Authors · 2024-11-27
> > **Looking Forward to Your Feedback**
> >
> > Dear Reviewer,
> >
> > This is a kind reminder regarding our paper. We sincerely appreciate your thoughtful review and the time you dedicated to it. We understand that you may have a busy schedule with many papers to review, but we would greatly appreciate it if you could take the time to consider our response.
> >
> > We have made significant efforts to address your concerns and would value your feedback on whether our responses effectively resolve them. If further clarification is needed, we would be happy to provide it. Your feedback is crucial for improving the quality of our paper, and your support of our work would be deeply appreciated.
> >
> > Thank you for your time and consideration.

---

### Official Review · Reviewer_4XM8 · 2024-10-31

**Soundness:** 3
**Presentation:** 4
**Contribution:** 3
**Rating:** 6
**Confidence:** 3

**Summary:**

This paper introduces a new SFT loss, GEM, based on the ideas of reverse KL loss and entropy. This loss can mitigate the overfitting issues of cross-entropy loss in the SFT process. Across a variety of task datasets, GEM demonstrates improved diversity and higher accuracy.

**Strengths:**

1. This article is well-written, with clearly organized content and significant research value.
2. The proposed GEM loss is simple and effective; this straightforward loss improvement simultaneously enhances diversity and accuracy.
3. The experiments are thorough, with validation across various mainstream tasks for large models.

**Weaknesses:**

1. Compared to CE Loss, GEM may introduce additional hyperparameters, which could make training more challenging.
2. There is no comparison of training costs across different methods.

**Questions:**

1. I understand that the introduced GEM can make the model's output distribution more diverse; however, increased diversity usually results in lower accuracy on domain-specific datasets or a higher probability of hallucination errors. Why does GEM improve both generalization and accuracy simultaneously?

2. What is the function of h in Equation 2? Is h necessary? If not, what would happen if h were omitted? And if h is optional, why are only the two variants mentioned in the paper permissible?

3. What is the functional difference between reverse KL and normal KL?

---

> ### Author Response · Authors · 2024-11-21
> **Response (Part I)**
>
> Thank you for taking the time to review our paper and provide insightful feedback. We address your concerns and answer your questions in the following part.
>
> **Comment 1:** GEM introduce additional hyper-parameters, which could make training more challenging.
>
> **Response 1:**  We would like to clarify that GEM introduces only a single hyperparameter, $\beta$, for regularization, similar to other baseline methods that also involve one hyperparameter for regularization. Furthermore, GEM's performance is robust to this hyperparameter, as shown in Figure 8 of Appendix F.3 (also provided in the anonymous link: https://anonymous.4open.science/r/iclr2025_submission_5769-2E96/images/figure_8.png). Additionally, using the techniques proposed in Section 4.2, we did not encounter any training stability issues compared with the standard CE-based method. In practice, we have successfully applied GEM to train 70B models, demonstrating its scalability.
>
> **Comment 2:** There is no comparison of training costs across different methods
>
> **Response 2:**  Thank you for pointing this out. In practice, we observed that GPU memory consumption and training speed are nearly identical across all methods. Detailed training costs of all method are reported below.
>
> |               | CE    | CE+WD | CE+Entropy | NEFT | GEM-Linear | GEM-Logsigmoid |
> | ------------- | ----- | ----- | ---------- | ---- | ---------- | -------------- |
> | Training Time | 5h58m | 6h1m  | 5h57m      | 6h8m | 5h55m      | 6h2m           |
>
> We note that all methods rely on the model's distribution for learning, allowing the loss to be directly calculated after a single forward pass through the Transformer. Thus, there is no significant difference in computation speed. For further details, please refer to Appendix B.
>
> **Question 3:** Increased diversity usually results in lower accuracy on domain-specific datasets or a higher probability of hallucination errors. Why does GEM improve both generalization and accuracy simultaneously?
>
> **Response 3:** Thank you for your thoughtful question. It seems there might be a typo in your question. We assume you are asking why GEM improves both generalization and diversity. Please feel free to correct us if we misunderstood.
>
> We believe the intuition that increased diversity typically reduces accuracy stems from two observations:
>
> 1. **Tasks with unique correct prediction:** For tasks where a single correct answer exists, increasing diversity in the output distribution can intuitively lower the probability of predicting the correct answer, leading to reduced performance.
> 2. **Temperature adjustment:** Practitioners often increase the temperature of the softmax distribution during inference to boost diversity, which is frequently observed to reduce accuracy (we do observe this; see Appendix F.4).
>
> Please tell us if you have additional insights. Below, we explain how GEM addresses these concerns and achieves its improvements:
>
> 1. **Open-ended tasks and response diversity:**  Many generative tasks, especially in GPT contexts, allow multiple valid responses. For instance, math reasoning tasks may follow different paths to the same answer, and conversational tasks can vary stylistically while remaining valid. Our study recognizes this: SFT training data often lack diverse responses, and CE loss amplifies overfitting, reducing diversity. To tackle this challenge, GEM is designed to mitigate over-memorization of the training data while encouraging the generation of diverse responses. We will explain this further below.
>
> 1. **Training diversity vs. inference tricks:** The diversity-accuracy trade-off often observed with temperature scaling arises because this technique is applied during **inference** without grounding in data information. It amplifies low-probability regions of the model's distribution for increasing diversity, increasing error rates. In contrast, GEM introduces diversity at the **training** phase by aligning the model's distribution with the data distribution. This approach ensures that diversity is meaningful and grounded in the data.
> 2. **Theoretical foundation:** GEM's effectiveness is rooted in reverse KL minimization with entropy regularization. These objectives are defined directly across the model's distribution, making the learning process **generative**, unlike the CE method, which operates over a fixed distribution. This generative approach focuses learning on the **high-probability regions** of the data distribution (see analysis in Section 4.2). By doing so, GEM captures genuine diversity by identifying multiple correct responses, rather than artificially boosting diversity with incorrect outputs.
>
> In summary, by leveraging training data and a carefully designed learning objective, GEM enhances both generalization and accuracy. This data-driven, generative learning paradigm ensures that diversity is introduced meaningfully rather than through heuristic adjustments.

---

> ### Author Response · Authors · 2024-11-21
> **Response (Part II)**
>
> **Question 4:** What is the function of $h$ in Equation 2? Is $h$ is necessary? If not, what would happen if $h$ were omitted? And if $h$ is optional, why are only the two variants mentioned in the paper permissible?
>
> **Response 4:** The function $h$ primarily shapes the loss landscape and is designed to increase training stability. While omitting $h$ is theoretically feasible, it may introduce practical challenges in optimization. Specifically, if $q$ is fixed and $h$ is omitted (i.e., $h$ is a linear function), optimizing $f$ in Equation 2 could result in unbounded outputs of $f$. To address this, it is preferable to choose a function like logsigmoid, which imposes an upper bound and helps stabilize optimization. This approach is also supported by prior work [1]. In our setting, the practical benefits of using such a function are demonstrated in Figure 2.
>
> We acknowledge that other alternatives for designing $h$ may exist, as discussed in [2]. Thus, $h$ is kept in our framework to allow flexibility and future extensions.
>
> [1] Zixiang Chen, Yihe Deng, Huizhuo Yuan, Kaixuan Ji, and Quanquan Gu. Self-play fine-tuning converts weak language models to strong language models. arXiv preprint arXiv:2401.01335, 2024.
>
> [2] Alexia Jolicoeur-Martineau. On relativistic f-divergences. In International Conference on Machine Learning, pp. 4931–4939. PMLR, 2020.
>
>
> **Question 5:** What is the functional difference between reverse KL and normal KL (i.e., the forward KL)?
>
> **Response 5:** Reverse KL and normal KL differ in both theory and practical application. Below, we clarify these differences:
>
> First, consider two distributions: $p$ (the data distribution) and $f$ (the model distribution). The two divergences are defined as follows:
> $$
> D_{\operatorname{normal kl}}(p, f) = \sum_{x} p(x) \log \frac{p(x)}{{f(x)}}  = -H(p) - \sum_{x}p(x) \log f(x)
> $$
>
> $$
>        D_{\operatorname{reverse kl}}(p, f) = \sum_{x} f(x) \log \frac{f(x)}{{p(x)}} = - H(f) - \sum_{x} f(x) \log p(x)
> $$
>
> Both two divergences define a "distance" measure for two distributions. The normal KL is defined with respect to the distribution $p$ while the reverse KL is defined with respect to the distribution $f$. In theory, we have that $p=f$ if and only if $D=0$ for both divergences. However, in practice, when finite samples are used to approximate the expectation, significant differences arise:
>
> - Normal KL Divergence: It uses samples from the fixed, offline data distribution $p$. This allows a direct sample approximation of the expectation. But, since the training is offline, this approach can result in distribution shifts during inference [1], leading to issues such as the well-known teacher-forcing problem [2].
> - Reverse KL Divergence: It computes the expectation over the model’s distribution $f$, which avoids issues like distribution shift since it inherently considers the model’s generative process. However, reverse KL formulations are challenging to solve in practice because $p(x)$ is typically unknown, making it difficult to estimate $\log p(x)$ and calcaulte the loss.
>
>
> When combined with an entropy regularizer in distribution matching, the normal and reverse KL divergences exhibit additional differences:
>
> - Normal KL Divergence:  The forward KL is defined with respect to the data distribution $p$, while the entropy regularizer is defined with respect to the model's distribution $f$. This setup tends to amplify the tail probabilities when increasing diversity (see our gradient analysis in Appendix D.2 for details).
> - Reverse KL Divergence: Both the reverse KL and the entropy regularizer are defined with respect to the model's distribution $f$, which offers several advantageous properties (see Section 4.2). In this work, we propose a new approach for solving this problem, even though the reverse KL cannot be directly computed due to the unknown $p(x)$.
>
> Please let us know if this explanation addresses your question, or if further clarification is needed—we would be happy to provide additional details.
>
> [1] Ross, Stéphane, Geoffrey Gordon, and Drew Bagnell. "A reduction of imitation learning and structured prediction to no-regret online learning." Proceedings of the fourteenth international conference on artificial intelligence and statistics. JMLR Workshop and Conference Proceedings, 2011.
>
> [2] Williams, Ronald J., and David Zipser. "A learning algorithm for continually running fully recurrent neural networks." Neural computation 1.2 (1989): 270-280.
>
>
> ----
>
> We sincerely thank you for your thoughtful review. We hope our responses have adequately addressed your concerns, and we are happy to provide further clarification if needed. We would greatly appreciate it if you could re-evaluate our paper and consider adjusting your recommendation accordingly.

---

> > ### Comment · Reviewer_4XM8 · 2024-11-26
> > **Thank you for the detailed response**
> >
> > Thank you for the detailed response. I decided to keep my ratings.

---

> > > ### Author Response · Authors · 2024-11-27
> > > **Thanks for Your Feedback**
> > >
> > > We are happy that our response has addressed your concerns and sincerely thank you for your valuable feedback.

---

### Official Review · Reviewer_yyC9 · 2024-11-02

**Soundness:** 3
**Presentation:** 3
**Contribution:** 3
**Rating:** 6
**Confidence:** 2

**Summary:**

The use of cross-entropy (CE) loss during the supervised fine-tuning stage often results in overfitting and a reduction in output diversity. To address this limitation, this work introduces a novel training method called GEM. GEM tackles the challenge of distribution matching through reverse KL divergence minimization and incorporates maximum entropy regularization to promote output diversity. The authors demonstrated the effectiveness of GEM across multiple scenarios, showcasing its ability to maintain high performance while reducing overfitting. Additionally, they highlighted that GEM offers benefits for test-time computation, where maintaining diversity is crucial.

**Strengths:**

1. The work is well-motivated, and the writing is clear.
2. They demonstrated increased diversity in the model's output through high performance on the test-time compute set.
3. They designed a method that is tractable for sequential data.

**Weaknesses:**

1. The goal seems to be heavily focused on enhancing the diversity of generations. I believe it would be beneficial to further evaluate how the proposed method performs on tasks where diversity is not as critical. For instance, it would be helpful to show how it performs on tasks where factual accuracy is important.

**Questions:**

1. This is not a critical question, but is there any method to demonstrate that the performance improvement in test-time computation truly results from the increased diversity in the model's output?
2. Is there any performance degradation when increasing the temperature at inference time?
3. Could you also show performance results using LLM-as-a-judge in the Creative Writing section? While diversity is an important dimension, I believe it is also necessary to evaluate other aspects such as the coherency of the writing.

---

> ### Author Response · Authors · 2024-11-21
> **Response (Part I)**
>
> Thank you for taking the time to review our paper and provide valuable feedback. We greatly appreciate your efforts and have addressed your concerns and questions below.
>
> **Comment 1:** The goal seems to be heavily focused on enhancing the diversity of generations. It would be helpful to show how it performs on tasks where factual accuracy is important.
>
> **Response 1:** Thank you for your insightful suggestion. We agree that both factual accuracy and diversity are essential aspects for a model to perform well in specific tasks. While GEM aims to increase diversity and avoid overfitting, we also extensively evaluate its performance on tasks where factual accuracy is critical. For example:
>
> - **Section 5.1:** In general instruction-tuning tasks, GEM reduces perplexity by 9% relative (see Table 5 in the Appendix), highlighting its effectiveness in modeling data distributions. For instruction-following evaluations, GEM achieves a 4% relative improvement over CE on IF-Eval when using greedy decoding. This evaluation prioritizes accuracy over diversity, as it often imposes strict output constraints. Likewise, on math reasoning tasks such as GSM8K, where the correctness of reasoning steps is critical, GEM demonstrates an 11% relative improvement over CE, also with greedy decoding.
> - **Section 5.2:** In domain-specific fine-tuning, GEM delivers significant gains, achieving an 8-point relative improvement on competition-level MATH tasks and a 12% relative improvement on HumanEval code generation tasks, both evaluated deterministically using greedy decoding.
>
> **Question 2:** This is not a critical question, but is there any method to demonstrate the performance improvement in test-time compute truly results from the increased diversity in the model's output?
>
> **Response 2:** A common approach to demonstrating the benefits of increased diversity is to evaluate performance scaling relative to the generation budget during inference. When given the same sampling budget, models that produce more diverse outputs typically achieve better performance by identifying better solutions. Conversely, if a model requires a smaller sampling budget to achieve comparable performance, it indicates reduced redundancy in its outputs, reflecting higher diversity.
>
> In our experiments, when using the same sampling budget, GEM achieves better performance, with up to a 5-point improvement in math reasoning and an 8-point improvement in code generation, as demonstrated in Section 5.1. Furthermore, it consistently requires a significantly smaller sampling budget—often 50% less—when aiming to achieve comparable performance with the baselines. This efficiency highlights the role of enhanced output diversity. For further details, please refer to the newly added Figure 3 in Section 5.1 (also provided in the anonymous link: https://anonymous.4open.science/r/iclr2025_submission_5769-2E96/images/figure_3.png) and the comprehensive evaluation in Appendix F.
>
> **Question 3:** Is there any performance degradation when increasing the temperature at inference time?
>
> **Response:** Yes, we observed performance degradation at higher temperatures. For instance, consider the GSM8K math reasoning task using models fine-tuned with the UltraFeedback dataset. CE's performance decreased from 46.9 to 45.8, and GEM's performance dropped from 50.9 to 48.6 when the temperature increased from 0.6 to 0.8 with random sampling. These results are detailed in Figure 9 in Appendix F.4 (also provided in the anonymous link: https://anonymous.4open.science/r/iclr2025_submission_5769-2E96/images/figure_9.png).
>
> This degradation likely occurs because higher temperatures increase the tail probabilities of the output distribution, raising the risk of errors. This highlights the limitations of temperature adjustments for improving diversity. In contrast, our approach promotes diversity during training by leveraging supervision signals from the data, enabling a more principled and effective enhancement of output diversity.

---

> ### Author Response · Authors · 2024-11-21
> **Response (Part II)**
>
> **Question 4:** Could you also show performance results using LLM-as-a-judge in the Creative Writing section?
>
> **Response**:  Certainly! Using the evaluation protocol in [1], we employed LLM-as-a-judge to evaluate writing quality. Our results show that GEM improves writing quality by approximately 4% compared with CE and outperforms other baselines.
>
> Evaluation of writing quality for story writing (higher scores indicate better performance; maximum score: 10):
>
> |                          | CE   | CE+WD | CE+Entropy | NEFT | GEM-Linear | GEM-Logsigmoid |
> | ------------------------ | ---- | ----- | ---------- | ---- | ---------- | -------------- |
> | Story Wrting (avg score) | 6.90 | 6.86  | 7.08       | 6.89 | 7.18       | **7.20**       |
> | Story Writng (max score) | 8.17 | 8.16  | 8.29       | 8.16 | **8.40**   | 8.39           |
>
>
> Evaluation of writing quality for poem writing (higher scores indicate better performance; maximum score: 10):
>
> |                         | CE   | CE+WD | CE+Entropy | NEFT | GEM-Linear | GEM-Logsigmoid |
> | ----------------------- | ---- | ----- | ---------- | ---- | ---------- | -------------- |
> | Poem Wrting (avg score) | 6.87 | 6.83  | 7.06       | 6.88 | 7.17       | **7.21**       |
> | Poem Writng (max score) | 8.12 | 8.11  | 8.31       | 8.12 | 8.42       | **8.43**       |
>
>
> Details: The LLM judge model assessed responses on five criteria—helpfulness, relevance, accuracy, depth, and creativity—using the evaluation prompts from FastChat's judge prompts (https://github.com/lm-sys/FastChat/blob/main/fastchat/llm_judge/data/judge_prompts.jsonl). Each response was scored on a scale from 1 to 10. The evaluation involved 500 questions each for poem and story writing, with 16 responses per question, resulting in a total of 96,000 responses across 6 methods.
>
> To compute the scores, we calculated both the average and maximum scores across the 16 responses for each question, then averaged these scores across all 1,000 questions. To reduce evaluation costs, we used the open-source Llama-3.1-70B-Instruct model, a strong LLM well-suited for assessing responses.
>
> [1] Zheng, Lianmin, et al. "Judging llm-as-a-judge with mt-bench and chatbot arena." Advances in Neural Information Processing Systems 36 (2023): 46595-46623.
>
> ---
>
> We sincerely thank you for your thoughtful review, comments, and questions. We gratefully hope that you could re-evaluate our paper based on the revised manuscript and the clarifications provided above. If our responses have satisfactorily addressed your concerns, we would greatly appreciate it if you could consider updating your review score accordingly. However, if you have any additional concerns, please do not hesitate to let us know. We are more than willing to provide further clarification.

---

> > ### Author Response · Authors · 2024-11-27
> > **Looking Forward to Your Feedback**
> >
> > Dear Reviewer,
> >
> > This is a kind reminder regarding our paper. We understand that your schedule is likely very busy, with many papers to review, but we sincerely look forward to hearing your feedback.
> >
> > We have put many efforts to address your concerns and would greatly appreciate your feedback on whether our responses effectively resolve them. If further clarification is needed, we would be more than happy to provide it. Your feedback and support are valuable to us and deeply appreciated.
> >
> > Thank you for your time and attention.

---

### Official Review · Reviewer_xjk9 · 2024-11-02

**Soundness:** 3
**Presentation:** 3
**Contribution:** 3
**Rating:** 8
**Confidence:** 3

**Summary:**

The authors address the overfitting and decrease in output diversity that arise when using the cross-entropy loss during supervised fine-tuning by introducing a novel distribution matching method called *Generative and Entropy-regularized Matching of distributions* (GEM). GEM is based on two core principles: (1) the model should assign higher probabilities to observed data without overtraining, and (2) as supervised datasets cannot encompass all possible data and thus only cover a limited distribution, the model should learn from both the supervised data and its own generated samples. The first principle is addressed by maximizing the entropy, while the second is achieved by minimizing the reverse KL divergence between the model and target distributions. The authors showed that GEM effectively improves the output diversity, mitigates overfitting, and outperforms the cross-entropy loss (with and without weight decay and entropy regularization) as well as NEFT on multiple benchmarks, including IFEval, HumanEval, GSM8K, MATH, and MBPP.

**Strengths:**

- SFT is a timely and impactful area of research as it is an essential phase of the LLM pipeline.
- The proposed GEM method is grounded in theory, effectively tackles overfitting, and improves the output diversity of Llama3-8B.
- The experimental setup is modern.
- The paper is clear and well-structured.

**Weaknesses:**

- GEM is only evaluated on Llama3-8B, which might particularly benefit from GEM.
- While Best-Of-N (BON) is useful to show the diversity of the output, it is not a practical solution and therefore does not represent the performance of the model.

**Questions:**

- In addition to Majority Voting (MV) and Best-Of-N (BON), can you report the performance using an LLM as a judge on the same 32 samples?
- Out of the 32 generated samples, how prevalent is the selected answer by MV? In other words, can you report the ratio of the selected answers when using majority voting?
- For the CE + entropy baseline [1], what coefficient γ did you use to weight the regularization? Did you conduct a grid search?
- Line 33: I suggest replacing "Despite extensive pre-trained, " with "Despite extensive pre-training, " or "Despite being extensively pre-trained, ".
- Line 423: Entropy regularization supports Principle 1.

[1] Abhimanyu Dubey, Otkrist Gupta, Ramesh Raskar, and Nikhil Naik. Maximum-entropy fine grained classification. Advances in neural information processing systems, 31, 2018.

---

> ### Author Response · Authors · 2024-11-21
> **Response (Part I)**
>
> Thank you for taking the time to read our paper and for providing a positive review of our work. We appreciate your thoughtful feedback. Below, we address your concerns and answer  your questions.
>
> **Comment 1:** GEM is only evaluated on Llama3-8B, which might particularly benefit from GEM.
>
> **Response 1:** Thank you for raising this concern. We selected the Llama-3-8B model for our study because it was one of the most strong 8B-scale models available at the time of our research. To address your feedback, we extended our experiments to include models ranging from 3B to 70B, such as Qwen-2.5-3B, Qwen-2.5-7B, Gemma-2-9B, and Llama-3.1-70B. These models are state-of-the-art and were released in recent months.
>
> The results of these experiments are shown in Figure 11 in Appendix F, which is also accessible via the anonymous link: https://anonymous.4open.science/r/iclr2025_submission_5769-2E96/images/figure_11.png. Our findings indicate that baselines such as weight decay and NEFT often underperform relative to CE, while CE+Entropy produces results comparable to CE. Notably, GEM consistently demonstrates substantial improvements over CE across all evaluated model sizes. These results provide strong evidence that GEM is not only well-suited for Llama-3-8B but also other models.
>
> **Comment 2**: While Best-Of-N (BON) is useful to show the diversity of the output, it is not a practical solution and therefore does not represent the performance of the model.
>
> **Response 2**: Thank you for raising this concern. While we recognize its computational limitations, we respectfully argue that BON remains a valuable metric for assessing model performance. It has been successfully utilized in prior research and holds promise for future applications. As a training-free method, BON is widely employed to estimate a model's potential for further RLHF training [1, 2] or self-improvement through distillation [3]. Additionally, it has proven effective in scenarios such as test-time compute scaling [4].
>
> We would like to note the growing interest in developing accelerated BON sampling techniques [5], which directly address some of the computational challenges you highlighted.
>
> [1] Stiennon, Nisan, et al. "Learning to summarize with human feedback." Advances in Neural Information Processing Systems 33 (2020): 3008-3021.
>
> [2] Gao, Leo, John Schulman, and Jacob Hilton. "Scaling laws for reward model overoptimization." International Conference on Machine Learning. PMLR, 2023.
>
> [3] Sessa, Pier Giuseppe, et al. "Bond: Aligning llms with best-of-n distillation." arXiv preprint arXiv:2407.14622 (2024).
>
> [4] Snell, Charlie, et al. "Scaling llm test-time compute optimally can be more effective than scaling model parameters." arXiv preprint arXiv:2408.03314 (2024).
>
> [5] Sun, Hanshi, et al. "Fast Best-of-N Decoding via Speculative Rejection." arXiv preprint arXiv:2410.20290 (2024).
>
> **Comment 3:** In addition to Majority Voting (MV) and Best-Of-N (BON), can you report the performance using an LLM as a judge on the same 32 samples?
>
> **Response 3:** Certainly. We have applied the LLM-as-a-judge method from [1] to evaluate response quality. For each question, we have calculated the average score and best score across 32 samples. The averaged results, with a maximum score of 10, demonstrate that GEM outperforms CE in this evaluation:
>
> |               | CE   | CE+WD | CE+Entropy | NEFT | GEM-Linear | GEM-LS |
> | ------------- | ---- | ----- | ---------- | ---- | ---------- | ------ |
> | Average Score | 7.09 | 7.12  | 7.05       | 7.10  | 7.14       | **7.15**   |
> | Best Score    | 8.36 | 8.43  | 8.41       | 8.38 | **8.47**       | 8.46   |
>
> Details: The LLM judge model assessed responses on five criteria—helpfulness, relevance, accuracy, depth, and creativity—using the evaluation prompts from FastChat's judge prompts (https://github.com/lm-sys/FastChat/blob/main/fastchat/llm_judge/data/judge_prompts.jsonl). Each response was scored on a scale from 1 to 10. The evaluation involved 805 questions from AlpacaEval, with 32 responses per question, resulting in a total of 154,560 responses across 6 methods. To compute the scores, we calculated both the average and best scores across the 32 responses for each question, then averaged these scores across all 805 questions. To reduce evaluation costs, we employed the open-source Llama-3.1-70B-Instruct model as the judge.
>
> [1] Zheng, Lianmin, et al. "Judging llm-as-a-judge with mt-bench and chatbot arena." Advances in Neural Information Processing Systems 36 (2023): 46595-46623.

---

> ### Author Response · Authors · 2024-11-21
> **Response (Part II)**
>
> **Question 4:** Out of the 32 generated samples, how prevalent is the selected answer by MV? In other words, can you report the ratio of the selected answers when using majority voting?
>
> **Response 4**: Certainly. We understand that you are interested in the self-consistency of majority voting,  particularly as diversity increases. Below, we present the ratio of the majority-voted answer among the 32 generated samples, along with the majority voting accuracy for reference. These metrics were computed for each question and averaged across 1,319 questions from the GSM8K dataset.
>
> |                          | CE   | CE+WD | CE+Entropy | NEFT | GEM-Linear | GEM-LS |
> | ------------------------ | ---- | ----- | ---------- | ---- | ---------- | ------ |
> | Majority Voting Accuracy | 62.2 | 63.9  | 64.1       | 64.8 | 65.2       | 65.3   |
> | Ratio of Majority Vote   | 52.6 | 53.4  | 53.3       | 53.3 | 53.0       | 52.8   |
>
> From the second row of the table, we observe that the ratio of the majority-voted answer is consistently around 53% across all methods, with no significant differences. This result shows that GEM maintains self-consistency even as it increases diversity. This can be attributed to its theoretical foundation, which emphasizes learning diverse responses in the *mode probability* regions (see analysis in Section 4.2). By focusing on these regions, GEM captures multiple valid responses that reflect genuine diversity, rather than producing a mix of correct and incorrect responses to artificially boost diversity. This is the main punchline of GEM.
>
> **Question 5:** For the CE + entropy baseline, what coefficient $\gamma$ is used to weight the regularization? Did you conduct a grid search?
>
> **Response 5:** As detailed in Appendix E.1, we selected a coefficient of $\gamma = 0.1$, determined through a grid search over the range $[0.5, 0.1, 0.01, 0.001]$. Training with $\gamma = 0.5$ failed due to a significant increase in perplexity (15.3 compared to the typical value of 3.4). Among the remaining values, $\gamma = 0.1$ consistently achieved the best performance in both diversity and accuracy-focused tasks.
>
> Specifically, for output diversity, evaluated across three metrics for poem and story writing, the average scores were 51.2 ($\gamma = 0.1$), 48.0 ($\gamma = 0.01$), and 48.3 ($\gamma = 0.001$), with $\gamma = 0.1$ delivering the best performance. Similarly, for downstream accuracy on GSM8K using greedy decoding, $\gamma = 0.1$ achieved an accuracy of 45.94, surpassing 44.05 ($\gamma = 0.01$) and 43.52 ($\gamma = 0.001$). These results led us to select $\gamma = 0.1$ for our experiments.
>
> **Comment 6:** Line 33: I suggest replacing "Despite extensive pre-trained, " with "Despite extensive pre-training, " or "Despite being extensively pre-trained, ". Line 423: Entropy regularization supports Principle 1.
>
> **Response 6:** Thank you for your suggestions. We have made the corresponding revisions in the paper.
>
> ---
>
> We sincerely thank you for your thoughtful review. We gratefully hope that you could re-evaluate our paper based on the revised manuscript and the clarifications provided above. If our responses have satisfactorily addressed your concerns, we would greatly appreciate it if you could consider updating your review score accordingly. However, if you have any additional concerns, please do not hesitate to let us know. We are more than willing to provide further clarification.

---

> > ### Comment · Reviewer_xjk9 · 2024-11-23
> >
> > **Response 1**: Thank you for extending the evaluation to include Qwen-2.5-3B, Qwen-2.5-7B, Gemma-2-9B, and Llama-3.1-70B.
> >
> > **Responses 2 and 3**: Thank you the references and including LLM-as-a-judge.
> >
> > **Response 4**: I am glad to see that GEM maintains self-consistency.
> >
> > **Response 5**: Thank you for pointing out the hyperparameters I missed in the appendix and providing additional details.
> >
> > The authors have addressed all of my concerns and questions, and I have updated my score accordingly.

---

> > > ### Author Response · Authors · 2024-11-27
> > > **Thanks for your positive support**
> > >
> > > We are glad to see that our response effectively addresses your concerns. We sincerely thank you for your positive feedback and support!

---

### Author Response · Authors · 2024-11-21
**General Response and Summary of Changes**

We sincerely thank all reviewers and area chairs for their efforts in reviewing our paper and providing valuable feedback. We have carefully addressed each concern raised by the reviewers and made corresponding revisions to our paper. We highlight the revision in red. Below, we summarize the key changes in our revision:

- **Section 4.2 (Page 5):** We emphasize the computational advantages of GEM, including *single-model optimization* and *accurate gradient estimation*. This approach allows GEM to effectively solve the challenging reverse-KL-based distribution matching problem and scale efficiently to models with 70B parameters, for which prior GAN-style algorithms may struggle to solve effectively.
- **Section 5.1 (Page 9):** We visualize GEM's test-time scaling performance across different generation budgets, showcasing the benefits of its enhanced diversity. Notably, to match the performance of baselines, GEM often requires only 0.5x the sampling budget.
- **Section 5.2 (Page 10):** We discuss how GEM enhances both diversity and generalization. Additionally, Appendices D.1 and D.2 have been revised to explain why the classical weight decay technique and the naive CE approach with entropy regularization often fall short in achieving these improvements.
- **Appendix B (Page 16 - Page 17):** We detail the implementation of GEM and analyze its computational efficiency, demonstrating that it matches the computation costs of CE.
- **Appendix F.1 (Page 24 - Page 25):**
    - *Quality of Creative Writing:* We use LLM-as-a-judge to evaluate the quality of responses in writing poems and stories. GEM improves both diversity and overall response quality.
    - *Quality of Chatting Responses:* For the chatting task, we employ LLM-as-a-judge to evaluate response quality. GEM outperforms the baselines, largely due to its ability to mitigate overfitting.
    - *Self-Consistency Analysis:* We analyze self-consistency in majority voting by calculating the ratio of majority votes. The results show that GEM achieves high self-consistency across diverse responses, consistent with its generative learning mechanism.
- **Appendix F.3 (Page 27):** We perform a sensitivity analysis of GEM's hyper-parameter to assess their impact. The results demonstrate that GEM is robust to the hyper-parameter.
- **Appendix F.4 (Page 27 - Page 28):** We show the limitations of temperature adjustment as an ad-hoc inference-stage trick for increasing output diversity. Furthermore, we demonstrate that by incorporating diversity into the training process, GEM achieves better quality and diversity compared than this trick.
- **Appendix F.5 (Page 28)**: We show that GEM outperforms CE consistently in low-data regimes.
- **Appendix F.6 (Page 28):** We provide 24 new evaluation results comparing baseline methods and our approach across models ranging from 3B to 70B parameters, including Qwen-2.5-3B, Qwen-2.5-7B, Gemma-2-9B, and Llama-3.1-70B. The findings highlight that GEM consistently outperforms CE, while baseline methods often fall short of delivering improvements over CE.

We believe that the revisions outlined above have improved the quality of our paper, thanks to the insightful comments from the reviewers. We hope these new results can address the reviewers' concerns and further strengthen the contributions of our work. Thank you once again for your valuable feedback.

---

### Author Response · Authors · 2024-12-04
**Summary of Rebuttal Discussion**

Dear Reviewers, ACs, SACs, and PCs,

We sincerely thank you for your efforts in reviewing our paper! We would like to take this opportunity to summarize the rebuttal discussion to facilitate further discussion among reviewers and ACs.

We received four review comments, most of which recognize the soundness, presentation, and contribution of our paper, with evaluations ranging from "good" to "excellent". We are pleased that our responses successfully addressed the concerns raised by Reviewer 4XM8 and Reviewer xjk9. Although we did not receive follow-up feedback from Reviewer yyC9 and Reviewer wHN6, we greatly appreciate their initial comments and have provided detailed responses to address potential misunderstandings. Below, we summarize their main concerns and our responses:

- Reviewer yyC9 recommended evaluating factual accuracy alongside diversity. In response, we provided evidence of our method's performance on instruction-following and math reasoning tasks under greedy decoding, where there is no randomness and the focus is on factual accuracy. Reviewer yyC9 also proposed the use of LLM-as-a-Judge. To address this, we employed the Llama-3.1-70B-Instruct model to evaluate 96,000 responses, demonstrating that GEM not only improves diversity but also enhances response quality, as claimed. We believe these results address Reviewer yyC9's concerns effectively.
- Reviewer wHN6 raised questions about the scalability of GEM to other architectures and its performance on larger models. This is an important point, and we have tested GEM on models with up to 70B parameters across various modern architectures, including Qwen, Llama, and Gemma. Our experiments confirm that GEM performs well across architectures, further validating its robustness. Importantly, GEM operates at the same computational speed as CE, making it a reliable and practical method for practitioners. Additionally, as clarified in our response to Reviewer wHN6, our proposed techniques are new and not covered in prior literature, underscoring the technical contribution of our work.


All review comments have helped us improve the quality of our paper. We have provided over 3 technical discussions and clarifications and presented more than 10 experiment results in the revision (detailed in the "General Response and Summary of Changes" section). These revisions further emphasize our paper's three key contributions:

1. We introduce the framework of entropic distribution matching for SFT of LLMs, addressing the limitations of overfitting and distribution collapse associated with the default CE method.
2. We design a new training method, GEM, that addresses the training challenges of reverse KL divergence minimization and maximum entropy regularization within our framework. Furthermore, GEM is supported by a theoretical guarantee of convergence.
3. Across models ranging from 3B to 70B, we empirically demonstrate that GEM mitigates overfitting of training data and improves the diversity of responses, leading to increased efficiency in test-time scaling.

In summary, we believe our work presents a new framework for fine-tuning LLMs, with techniques that may also be valuable for other directions, such as self-improvement through best-of-n distillation, addressing preference collapse in RLHF, and more (as detailed in Appendix G of our paper).

Thank you again for your time and consideration! We hope the above clarifications are helpful.

Sincerely,

The Authors

---

### Meta-Review · Area_Chair_VoQa · 2024-12-23

**Metareview:**

This paper proposes a new distribution matching method called Generative and Entropy-regularized Matching of distributions (GEM), which is based on reverse KL divergence and entropy. It aims to address the issues of overfitting and reduced output diversity that often occur when using cross-entropy (CE) loss in the SFT process. The motivation is clear, the expression is well-articulated, and the experiments are thorough and comprehensive. Although the reviewer raised concerns regarding training costs and the generalizability of the experimental models, the authors conducted thorough experiments during the rebuttal and effectively addressed the issues. Overall, this is a solid work, so the AC recommends accept.

**Additional Comments On Reviewer Discussion:**

The main concerns raised by the reviewers were the additional cost introduced by the new method (from reviewers 4XM8 and wHN6) and the scalability to other architectures (from reviewers xjk9 and wHN6). The authors addressed these issues by adding more experiments and providing detailed explanations and analyses.

---

### Decision · Program_Chairs · 2025-01-22

Accept (Poster)